

# Effects of changes in nutrient loading and composition on hypoxia dynamics and internal nutrient cycling of a stratified coastal lagoon

Yafei Zhu[1,2,*], Andrew McCowan[2], Perran L. M. Cook[1]

[1]Water Studies Centre, Monash University, Clayton, 3800, Australia
[2]Water Technology Pty Ltd, 15 Business Park Drive, Notting Hill, 3168, Australia

* *Correspondence to*: Yafei Zhu (Yafei.zhu@monash.edu)

**Abstract.** The effects of changes in catchment nutrient loading and composition on the phytoplankton dynamics,
development of hypoxia and internal nutrient dynamics in a stratified coastal lagoon system (the Gippsland Lakes) was
investigated using a 3D coupled hydrodynamic biogeochemical water quality model. The study showed that primary
productivity was equally sensitive to changed dissolved inorganic and particulate organic nitrogen loads, highlighting the
need for a better understanding of particulate organic matter bioavailability. Stratification and sediment carbon enrichment
are the main drivers for the hypoxia and subsequent sediment phosphorus release in the Lake King. High primary production
stimulated by large nitrogen loading brought by winter flood contributed almost all the sediment carbon deposition (as
opposed to catchment loads) which was ultimately responsible for summer bottom-water hypoxia. Interestingly, internal
recycling of phosphorus was more sensitive to changed nitrogen loads than total phosphorus loads, highlighting the potential
importance of nitrogen loads exerting a control over systems that become phosphorus limited (such as during summer
nitrogen-fixing blooms of cyanobacteria). Therefore, the current study highlighted the need to reduce both TN and TP for
water quality improvement in estuarine systems.



# 1 Introduction

Excessive anthropogenic nutrient loading, particularly nitrogen, has led to widespread hypoxia and other ecological damage in estuarine and coastal areas (Howarth et al., 2011). About half the known hypoxic events have been caused by eutrophication (Diaz and Rosenberg, 2008). High primary production as a result of eutrophication can lead to hypoxia or anoxia in poorly mixed bottom water and subsequently enhance the recycling of both nitrogen and phosphorus which again can reinforce eutrophication (Correll, 1998). This has been found in many stratified estuarine systems around the world including the Baltic Sea (Vahtera et al., 2007) and the Black Sea (Capet et al., 2016) , the Neuse River Estuary (Paerl et al., 1995) and the Gippsland Lakes (Scicluna et al., 2015). The magnitude of sediment phosphorus release is related to severity of bottom water dissolved oxygen (DO) depletion as well as the duration of hypoxia/anoxia. For example, Conley et al. (2002) found that the annual change in dissolved inorganic phosphorus (DIP) in Baltic Sea was proportional to the area covered by hypoxic water rather than the catchment phosphorus load.

Although some researchers argued that reduction in both nitrogen and phosphorus were important to improve hypoxia in areas such as the Gulf of Mexico (Rabalais et al., 2007) and the Baltic Sea (Vahtera et al., 2007) , others considered that nitrogen should be the primary factor driving marine coastal eutrophication (Diaz, 2001, Hagy et al., 2004, Howarth and Marino, 2006) and thus hypoxia. Regardless of this controversy, the global river export of phosphorus to the coastal ocean has already dropped significantly as a result of advances in wastewater treatment technology since the start of the 21st century; however, nitrogen export still remained high (Howarth et al., 2011). The form and composition of nitrogen export, i.e. dissolved inorganic nitrogen (DIN) and particulate organic nitrogen (PON), can also have significant impact on receiving coastal waters, as they have different residence time and bioavailability. Seitzinger et al. (2002) showed that total global PON and DIN export by rivers in 1990 are similar, but the DIN:PON ratios varied from region to region. Generally speaking, the DIN:PON ratio is much higher in areas with larger population indicating that anthropogenic activities had larger influence on DIN export compared to PON export. The global DIN inputs to coastal systems was predicted to increase by more than 120% to 47 million tN/year by 2050 compared to the level in 1990 (Seitzinger et al., 2002). The relative importance of internally generated (primary production stimulated by DIN export) and external supplied organic matter on hypoxia have only been studied by few researchers previously (Paerl et al., 1998). Most importantly, there was a lack of understanding of how different forms of nitrogen (i.e. DIN and PON) in catchment load can influence the dynamics of hypoxia and sediment phosphorus cycle in estuarine systems.

Coupled hydrodynamic and biogeochemical/ecological models are now increasingly sophisticated and can capture complex biogeochemical feedbacks. There have been a number of successful applications of these models in studying the effects of changes in anthropogenic nutrient loading on the water quality dynamics in receiving estuarine waters (Kiirikki et al., 2001, Webster et al., 2001, Neumann et al., 2002, Pitkänen et al., 2007, Skerratt et al., 2013). However, all these studies primarily focused on the effectiveness of alternative management scenarios for estuarine systems. None of these studies has addressed



the sensitivity of hypoxia dynamics and internal nutrient cycling to different forms of nitrogen, phosphorus and organic carbon inputs, which has important scientific and management implications for estuarine water quality.

In this study, we utilised a 3D coupled hydrodynamic-biogeochemical/ecological model to evaluate: 1) the sensitivity of phytoplankton and hypoxia dynamics in the Gippsland Lakes to the change in different compositions of anthropogenic nutrient loading, 2) and the consequent impact on internal nutrient dynamics.

## 2  Method

### 2.1 Study Site

The Gippsland Lakes, located in the southeast of Australia, are the largest estuarine coastal lagoon system in Australia (Figure 1). The system consists of three main lakes with total surface area of about 360 $km^2$.The depths vary from less than 4-m deep in Lake Wellington to 5-10m deep in Lake King. Lake Wellington and Lake Victoria are linked by McLennan Strait which is a narrow channel about 10 km long, 80 m wide and up to 11 m deep. The lakes are connected to the ocean through an artificial entrance at Lakes Entrance constructed in 1889. The Gippsland Lakes has a catchment area of more than 200,000 $km^2$ and mainly receives freshwater inflow from 6 major rivers as shown in Figure 1.

The Gippsland Lakes suffer recurring blooms of toxic Nitrogen-fixing cyanobacteria in summers following floods in winter and spring. Together with stratification, high carbon delivery to the sediment in winter and spring following flood-induced diatom/dinoflagellate blooms caused depleted bottom-water oxygen in summer, and a subsequent large release of phosphorus from the sediment in the central basins of Lake King and Victoria. Between July and August 2011, the Gippsland Lakes experienced two consecutive floods. In the following summer, a large toxic cyanobacteria bloom occurred in the lakes that persisted from mid-November 2011 to the end of January 2012. It was found that sediment phosphorus release as a result of depleted bottom DO rather than catchment load supplied most of the phosphorus to support the development of the bloom (Zhu et al., 2016).

### 2.2 Model Description

The coupled model used in the current study was developed by Zhu et al. (2016) using DHI's MIKE3 FM and ECOLAB. The horizontal domain of the hydrodynamic model was discretised as triangular and quadrilateral elements with element area ranging from 1,500 $m^2$ near the entrance channel and 3 $km^2$ in Bass Strait. The model had a total of 33 fixed z and varying sigma layers for the vertical discretization with height from less than 0.5 m close to the surface to 5 m down to the bottom. Smagorinsky formulation for the horizontal and the standard κ-epsilon model for the vertical have been used to simulate the



transport and turbulent mixing in the water column. A scaled eddy formulation was used for the horizontal and vertical dispersion processes which made the dispersion coefficients directly related to the eddy viscosity calculated by the turbulence models. With the same wind forcing data and model domain, a Spectral Wave model was also developed for the Gippsland Lakes, using DHI's MIKE 21 Spectral Wave module. The wave model results were used to calculate wind-wave

shear stress.

The water quality model contains 41 state variables describing the biogeochemical/ecological and chemical processes occurring in the water column and sediment compartments. The model included three groups of phytoplankton which were N-fixing cyanobacteria, vertically-migrating dinoflagellates and fast-growing diatoms. Organic carbon/nutrients were divided into labile and refractory fractions. To simplify, the dissolved organic carbon/nutrients and the hydrolysis process

(conversion from particulate to dissolved organic carbon/nutrient) have not been modelled explicitly. Instead, the model has been configured in the way that mineralisation of particulate organic carbon /nutrient took place without going through hydrolysis first.

It has previously been shown that internal phosphorus recycling is a key process within the Gippsland lakes requiring a refined implementation into water quality models (Webster et al., 2001). The present model overcame previous limitation by

implementing bioirrigation into the model enabling an accurate simulation of sediment phosphorus dynamics. The model also included a simple cohesive transport module which took into account of the shear stress due to wind-wave and current interaction.

The initial conditions, especially the sediment nutrient storage, could have a large impact on the model simulation. The

initial condition of organic carbon, nitrogen and phosphorus, and inorganic nitrogen in the sediment were estimated by iteratively simulating the model for a year and the concentration at the end of a simulation was used as the initial condition for successive simulations. This was repeated until the sediment nutrient inventory did not change substantially by the end of the simulation. A spatially varying sediment iron-bound phosphate distribution was estimated based on previous field studies.

The total sediment DIN, iron-bound phosphate, and labile particulate organic carbon (POC), PON and particulate organic phosphorus (POP) in the Gippsland Lakes were approximately 118 ton N, 3,234 ton P, 2,861 ton C, 238 ton N, and 34 ton P, respectively. The coupled model was calibrated and validated for the period between May 2010 and July 2012 and it has reproduced the hydrodynamic and biogeochemical conditions in the lakes well and successfully replicated the 2011-2012 summer cyanobacterial blooms.

**2.3 Nutrient scenarios**

The catchment nutrient load data were obtained from the Water Measurement Information System (previously was knowns as Victorian Water Resources Data Warehouse) which was managed by The Department of Environment, Land, Water and



Planning (DELWP). The nutrient data consisted of various constituent concentrations including total nitrogen (TN), nitrate, nitrite, ammonia, total Kjeldahl nitrogen (TKN), total phosphorus (TP) and DIP. The concentration of the total inorganic and particulate organic nutrients was first calculated using the raw data, and the particulate organic nutrients were then further divided into labile and refractory fractions. There was no measured data for the carbon input for the river flows so the

catchment organic carbon load was estimated using organic nitrogen load, assuming a C:N weight ratio of 10 (Meybeck, 1982) which also agrees with previous studies that showed that the sediment C: N weight ratio was around 8 to 13 in the Gippsland Lakes (Longmore, 2000, Holland et al., 2013). It can be assumed that organic matter with C:N ratio close to the Redfield ratio, 5.7 (on a mass basis), should be labile. The Redfield C:N ratio was close to 60% of that of the estimated catchment C:N ratio. Therefore, it was assumed that 60% of the catchment organic nutrients loads were labile and the rest

was refractory and not bioavailable over the timescale of water residence time. Therefore a very low mineralisation rate (0.005/day) was used for the refractory portion (Zhu et al., 2016).

The majority of the catchment nutrient loads are associated with flood events. On average, the western rivers (Latrobe, Thomson, and Avon River) and eastern rivers (Mitchell, Tambo and Nicholson River) each supply approximately 52% and

48% of the riverine freshwater inflows to the lake system. However, the western rivers contributed up to approximately 70% of catchment nutrient loads between May 2010 and July 2012 (Table 1). Majority of these nutrients were delivered during wet season and are associated with major floods (Figure 2). The DIN:PON ratio for the eastern rivers was only 0.6 compared to 1.17 for the western rivers, consistent with more intense human activities in the western catchment. In other words, the percentages of riverine bioavailable nitrogen (DIN+ labile PON) delivered to the Gippsland Lakes in the form of labile PON

were around 63% and 46% for the Eastern and Western rivers respectively . These numbers were very close to the ones reported for the other major rivers around the world, including the Mississippi (40%), Amazon (62%) and Yangtze (45%) Rivers (Mayer et al., 1998) and this confirmed that 60% of the particulate nutrient were labile was a valid assumption. The TN:TP ratios for both eastern and western rivers were around 10 (on a mass basis) which was close to the Redfield ratio of 7.23 (on a mass basis). While the DIN:DIP ratios were 22.3 and 18 (on a mass basis) for the eastern rivers and western rivers

respectively.

For the current study, we used the calibrated model as the base case and simulated a number of nutrient load scenarios for the same period. There were 5 sets of scenarios with adjusted loads for DIN, PON, TN (DIN+PON), TP (DIP + particulate P), or TN and TP. In all the scenarios, POC were set to vary the same proportion as PON. Since POC load was estimated

based PON load and the C:N ratio in the model was also used to define if particulate matter was labile or refractory. Each set of scenario had 8 simulations which decreased and increased the load by 25%, 50%, 75% and 100%. The response of bottom water DO and sediment processes to different nutrient scenarios were analysed and discussed, with the focus on the central basins of northern Lake King where the most severe hypoxia, highest sediment DIP fluxes and cyanobacterial blooms were located.





## 3 Results

### 3.1 Primary production

The annual total primary production (TPP) rate in Lake Wellington could reach as high as 600 g $C/m^2$/year, about 350 g

$C/m^2$/year in Lake Victoria followed by 250 g $C/m^2$/year in Lake King (Figure 3).  The spatial variation in primary production rate was caused by a number of factors, mainly including the higher nutrients loads from the western rivers and longer residence time for Lake Wellington. As a result, Lake King only contributed about 12% of the total primary production in the lakes. The total catchment POC load was only 7.5% of the TTP for the simulation period and was expected to have minor impact on the sediment biogeochemistry in the lakes.  As expected, the TPP in Lake King and the catchment

nutrient load had a positive correlation (Figure 4).  TPP was most sensitive to changes in TN+TP loads, and least sensitive to reductions in PON loads and increases in TP loads.  TPP was very sensitive to reductions in TP loads at reductions > 25%. The TN reduction scenarios displayed an intermediate response. TPP was more sensitive to reductions in TN than TP until reduction exceeded 75% when TPP became more sensitive to TP.

### 3.2 Bottom water oxygen

The total area in Lake King covered by hypoxic bottom water was about 40 $km^2$ for the base case (Figure 5). Any further increase in catchment DIN or TP load had no obvious impact on the total area of hypoxia, while slight increases were seen for the other three scenarios. This was because TPP did not increase much when either DIN or TP increased. Complete removal of catchment DIN and DON load would result in 25% and 50% reductions in the total area covered by hypoxic bottom water. The decrease in hypoxic area was insignificant when the TP load was reduced by 50% but followed by an

accelerating decline if TP was further reduced. On the other hand, the hypoxic area reduced more steadily when TN was reduced and decreased the most when TN and TP were reduced. The most severe and persistent hypoxia/anoxia was found in the northern Lake King basin. Therefore, the statistics of the bottom DO concentration time series from LKN (location shown in Figure 1) was extracted and analysed.  It has been found the bottom water DO concentration at LKN would still decrease to close to 0 mg/L even if either catchment TN or TP was completely removed (Figure 6). It would require

complete removal of both catchment TN and TP input to eliminate hypoxia in the northern Lake King basin. For the base case, the bottom DO concentration was below the 2mg/L threshold for almost 43% of the time (Figure 7) and the median bottom DO concentration was just slightly above 2mg/L (Figure 8). Compared to DIN and TP, doubling the PON load would result into much more frequent hypoxia at LKN, and the occurrence of hypoxia could substantially increase by almost 70% and also lead to 73% reduction in median bottom DO concentration. This was likely because the POC was also adjusted with

PON.  The median DO concentration would significantly increase to 5.5mg/L if catchment TN was completely removed. A



100% reduction in TP load would improve the median bottom DO concentrations to about 4 mg/L and reduction in PON would have a similar effect. Nonetheless, DIN tended to have a relatively smaller impact on the bottom DO concentrations at LKN.

### 3.3 Sediment Nutrient fluxes

The total CO2 (TCO$_2$) flux was a good indication of how much labile organic matter is deposited on the lake bed (Figure 9). To compare the effect of nutrient reductions on TCO2 fluxes, the effect of the initial sediment nutrient condition was taken into account by subtracting the TCO$_2$ flux and TPP for the simulation with no catchment nutrient input from all the model results. The TCO$_2$ flux and TPP were highly correlated (R$^2$=0.97, n=41) and TCO$_2$ flux was approximately equivalent to 8.5% of the TPP across the entire lake system. The ratio between TCO$_2$ flux and TPP at Lake King was higher than this at 33% (R$^2$=0.93, n=41). This indicated that the deposition rate in Lake King was much higher than rest of the lakes and/or the deposited organic matter could have come from the other parts of the lakes. Increased PON loads resulted in greater increase in TCO$_2$ flux than DIN or TP. The sediment ammonia fluxes followed very similar trends to the TCO$_2$ fluxes where ammonia fluxes were more dependent on PON load (Figure 10). The average annual nitrogen removal rate through denitrification was about 240 ton N/year in the sediment of Lake King for the base case. However, the nitrogen removal rate in Lake King for all the scenarios only varied marginally from the based case by around ±15%, except for the simulation of no anthropogenic nutrient load which had a slightly higher reduction in denitrification rate, approximately 35%. As a result, the denitrification efficiency in the sediment of Lake King had a negative correlation with the catchment nutrient load (Figure 11). Interestingly, sediment phosphate flux was more sensitive to the change in TN rather than TP loads (Figure 12). This was because majority of the phosphate fluxes was a consequence of desorption processes under hypoxic/anoxic conditions. The results also showed that even if the catchment nutrient load was completely removed, it would not stop sediment phosphate release and a 60% reduction was predicted. One explanation would be that desorption process sill took place as stratification prevented bottom-water oxygen replenishment after the flood event.

### 4 Discussion

Through observations and modelling, we have previously documented the seasonal dynamics of phytoplankton and nutrient cycling in the Gippsland lakes (Cook and Holland, 2012, Cook et al., 2010). High winter inflows carry nitrogen into the Gippsland Lakes which stimulate phytoplankton productivity. Inputs of organic matter from internal production and the catchment lead to hypoxia throughout spring and summer, which then drive phosphorus release from the sediment. We now discuss the sensitivity of this conceptual model to changes in external nutrient loading rates.

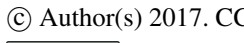


## 4.1 Primary production

Outside the summer cyanobacterial blooms, the lakes are typically nitrogen limited (Holland et al., 2012), and we therefore expected a strong sensitivity of primary production to nitrogen loading rates. Surprisingly, the model showed that primary production was equally sensitive to inorganic nitrogen loading and nitrogen loading in the form of particulate nitrogen and

that there were two distinct mechanisms by which these two nitrogen forms were trapped within the lakes. To calculate how much PON or DIN could have been retained in the lakes after the floods in July and August 2011, the model simulated the transport of PON and DIN excluding all the biological and chemical processes. The results from this simulation showed that 78.7% of flood-introduced PON was retained in the lakes by the end of August. Of the retained labile PON, 95% reached the sediment where roughly 90% was recycled over the following 3 months. The importance of PON as a nitrogen source

within aquatic systems will depend strongly on the degradation kinetics. In this study we estimated the degradation kinetics of PON based on a surprisingly small pool of literature (Cerco and Cole, 1994, Robson and Hamilton, 2004, DHI Water & Environment, 2012, Deltares, 2013). We therefore suggest further studies need to be undertaken to better understand the degradation kinetics of PON and the factors that control this such as land use.

On the other hand, without phytoplankton uptake, only 31.8% of the DIN remained in the lakes after the flood. In fact, all the simulations except for the TP reduction scenario showed that 70% ($R^2$=0.99, n=37) of catchment TN input remained in the lakes by the end of August 2011. It is suggested that majority of the DIN was assimilated by phytoplankton and transported to the sediment during the two-month high flow period. The corresponding TN export rate increased by 2.3%, 8.1%, 24% and 50% when TP was reduced by 25%, 50%, 75% and 100%, implying that the N:P ratio in flood waters can be an

important factor controlling the residence time of flood-introduced TN in the lakes. This has also been reflected in Figure 10 which shows a sharp decline in sediment ammonia flux when catchment TP was reduced, highlighting the importance of biogeochemical factors on the residence time of nutrients in estuaries (Church, 1986). Under the same hydrodynamic conditions, nitrogen in particulate form has a longer residence time as it can settle down to the bottom of the lake but DIN can be assimilated by phytoplankton and converted to PON before being washed out of the lakes. The rate and efficiency of

the conversion from DIN to PON ultimately determines the residence time of DIN in the system.

## 4.2 Hypoxia

Seasonal hypoxia is controlled by both stratification and inputs of organic carbon. The hypoxia observed in the Gippsland Lakes coincided with the recent transition to higher flow following the Millennium drought which ended in 2010. Boesch et al. (2001) also reported the extended hypoxia in the Chesapeake Bay in the 1970s which coincided with a transition from

drought to wet years. Large fresh water inflows do not only enhance stratification but also increase catchment organic carbon and nutrient load. Similar to many other estuaries systems such as the Baltic Sea (Conley et al., 2002) and the Black Sea (Capet et al., 2013), the supressed oxygen replenishment due to stratification and high oxygen consumption from the





mineralisation of deposited organic carbon were the main causes of the hypoxia in Lake King. Stratification in Lake King could last up to several months before the water column became well-mixed again. Lake King also had a higher deposition rate compared to the other parts of the lakes and the majority of the POC deposition was found to be in the deeper basin in northern Lake King. This was because a semi-closed circulation pattern was formed in this area as a result of the interaction

between the outgoing river flow and the incoming tidal flow from the ocean, resulting in a lot of POC being trapped in area unless there was a large flood or storm surge. Another important reason for POC retention in the Lake King basin was that the bottom shear stress in this area was generally very low, the normal current or wave conditions did not exert enough force to resuspend the sediment in this area once deposited. Furthermore, increases in nutrient load did not proportionally increase the total area subject to hypoxia in Lake King. This was because the areas subject to high detritus deposition were

determined by the hydrodynamics and wave conditions, and these depositional areas remained the same for all nutrient scenarios. Another important reason was that unlike the Baltic Sea, where low-DO water could move upwards from individual hypoxic basins and become connected to form a larger hypoxic region (Conley et al., 2009a), the Gippsland Lakes is a much shallower system and the bottom water in areas with depths < 3-4m were frequently ventilated with wind-driven upwelling and incoming tidal currents, even during the stratified period highlighting the importance of high organic

matter loads in the Gippsland Lakes maintaining hypoxia.

There has been some controversy as to whether primary production is simulated directly by anthropogenic nutrients or catchment organic carbon inputs stimulate increased nutrient recycling caused hypoxia in estuaries such as the Gulf of Mexico (Boesch et al., 2009). Previous studies have suggested that carbon either derived from algal blooms or the

catchment could result in estuarine hypoxia depending on the hydrological conditions (Paerl et al., 1998). To compare the relative importance of catchment carbon and primary production to the development of hypoxia in Lake King and sediment phosphorus flux, a mass balance calculation was undertaken to calculate amount of labile POC deposited in the sediment of Lake King. As the model in the current study was not able to trace the origin of the POC, the mass balance calculation was only an approximation and based on the difference between the base case and the simulation of complete removal of the

catchment TN and TP. For a given period, the change in sediment labile POC ($\Delta C$) is approximately equal to the sum of settled labile POC contributed by catchment input and primary production, subtracted by the sediment $CO_2$ flux due to mineralisation of labile POC. If the conversion from labile to refractory POC can be excluded from the calculation, then sediment $TCO_2$ flux for the zero catchment nutrient simulation could be assumed to be completely contributed by the mineralisation of refractory POC in the sediment (Cmr). The sediment $CO_2$ flux due to mineralisation of labile POC is equal

to sediment $TCO_2$ flux minus Cmr. Since the primary production for the zero catchment TP scenario between July 2011 and Jan 2012 was only equivalent to 0.38% of that for the base case, it could be assumed that all the sediment POC accumulation resulted from catchment input which is given by $\Delta C+TCO2- Cmr$. If applying the same approach to the base case, the total POC deposited in the sediment from both catchment and internal generation can be calculated.



The 2011 winter flood brought approximately 4,500 tons of labile organic carbon into the lakes and about 22% of this load settled in Lake King between July and August. For this period, catchment carbon contributed almost half of the sediment $TCO_2$ flux in Lake King. However, bottom-water hypoxia did not develop in Lake King until mid-October by which time, most of the sediment labile POC derived from the catchment was mineralised based on the results of the zero catchment TP

simulation. The elevated temperature and higher light levels during November resulted in higher primary production which consequently enhanced the sediment carbon enrichment, and increased the mineralisation rate of sediment POC. Severe bottom-water hypoxia developed and lasted through towards the end of the simulation. In addition, the catchment POC only contributed less than 7% the sediment $TCO_2$ flux between September 2011 and Jan 2012. This number is very close to the catchment POC to TPP ratio of 8.3% for the entire two-year simulation period. Therefore, bottom water depletion in Lake

King was primarily related to the planktonic organic matter stimulated by nutrient fluxes from the catchment.

### 4.3 Internal nutrient dynamics

Internal nutrient recycling can be a critical supply of nutrients to algal growth and it is therefore important to consider the sensitivity of these processes to changes in external loading. Denitrification efficiency is a commonly used measure of nitrogen removal efficiency from sediments (Eyre and Ferguson, 2009). Consistent with previous studies (Mulholland et al.,

2008, Gardner and McCarthy, 2009), denitrification efficiency increases with reduced nitrogen loading rates, which reduced sediment hypoxia and sediment organic carbon mineralisation rates. Interestingly, at high reductions in phosphorus loading there is also a large increase in denitrification efficiency, which results from the already noted transition to P limitation, meaning less organic nitrogen input to the sediment.

We have previously shown that the primary source of phosphorus fuelling summer blooms of nitrogen fixing bacteria is from

the sediment (Scicluna et al., 2015, Zhu et al., 2016). This sediment release of phosphorus is in turn induced by hypoxia, which is in turn driven by internal algal productivity. As already discussed, this algal productivity is driven by nitrogen during the winter and spring months and one would therefore expect sediment phosphorus release to be sensitive to nitrogen loads. Although, internal phosphorus loading could be potentially supressed by elevated nitrate concentration in the bottom water (Hemond and Lin, 2010), the model showed that the increase in oxidised depth of the sediment was limited due to

increased sediment oxygen demand and low diffusion rate of nitrate in the sediment. Of key importance, the model showed that internal phosphorus release from the sediment is more sensitive to total nitrogen loading than it is to total phosphorus loading at load reductions <80%. Conversely, increases in total phosphorus loads are expected to have a minimal effect on internal phosphorus recycling. There has previously been strong debate as to the importance of nitrogen versus phosphorus in coastal systems and it has been argued that the key focus of eutrophication management is to control phosphorus as it is

the nutrient that ultimately limits productivity (Schindler et al., 2008). Using a mechanistic approach, the present study highlights that both N and P reductions are required to reduce internal recycling of phosphorus, and that phosphorus load reductions alone are likely to be ineffective (Paerl, 2009, Conley et al., 2009b).




## 4.4 Management implications

Compared to DIN, PON had a slightly larger impact on the bottom DO concentration in Lake King most likely because POC load was related to PON load. The results showed that LKN was more susceptible to TN loading to a certain extent when compared to TP. However, initial input of catchment phosphorus was essential to stimulate primary production which contributed majority of the carbon enrichment in the sediment. To eliminate hypoxia in the Gippsland Lakes within the time scale (~2 years) of the current model simulation, would require a complete removal of catchment nutrient input. Many Studies have demonstrated the effectiveness of external nutrient reduction can be compromised by the sediment supply (Rossi and Premazzi, 1991, Istvánovics et al., 2002, Søndergaard et al., 2003, Jeppesen et al., 2005, Wu et al., 2017). In reality, it is very difficult to reduce nutrients sufficiently and for long enough. It has previously been estimated that the feasible reductions in TN and TP were only 25% (Ladson, 2012) and 20% (Roberts et al., 2012) respectively for the Gippsland Lakes. The results showed that even if the feasible reduction target was achieved, the immediate improvement in the water quality in the lakes was marginal. However, according to the experiences from pervious long-term studies of European lakes (Schindler and Hecky, 2009), it is possible that some effect of catchment nutrient reduction on bottom water DO might become obvious in Lake King after at least 5 to 10 years' continuous load control as sediment stores of phosphorus become run down.

For the Gippsland Lakes, majority of catchment nutrient flux was nonpoint source introduced by flood, making it difficult to manage. The reduction in dissolved inorganic nutrients in flood waters would be particularly more challenging. However, the current study has shown that the water quality in the lakes was also largely influenced by particulate nitrogen and phosphorus each of which comprised about 60% and 80% of the total catchment loads respectively between 2005 and 2011. Therefore, erosion control is the key to reduce particulate nutrient load from nonpoint source and improve the water quality in the Gippsland Lakes. Vegetated buffers, particularly riparian forests, have been considered to be the simplest but the most effective management option to reduce agricultural nonpoint source pollutants (Phillips, 1989), especially for nutrient in particulate forms. Vegetated buffer formed an important part of the water quality improvement strategies for many coastal and estuarine systems such as the Gulf of Mexico (Mitsch et al., 2001) and Chesapeake Bay (Lowrance et al., 1997). Zhang et al. (2010) analysed more than 50 published studies on the performance of vegetated buffers on nutrient removal and found that the median removal efficiency were 68.3% for nitrogen and 71.9% for phosphorus. However, the performance of vegetated buffers can be compromised at a catchment scale and the removal efficiency can be reduced dramatically to less than 20% (Verstraeten et al., 2006), due to sub-surface hydrological pathways, breakthrough surface runoff or bypass through roads (Mainstone and Parr, 2002). Other measures such as modification in agricultural practices should also be considered. Carefully designed and properly managed vegetated buffers can be a part to an integrated nonpoint source control strategy for estuarine water quality improvement.





## 5 Conclusion

Hypoxia and associated sediment phosphorus release in Lake King were predominantly driven by stratification and sediment carbon enrichment. Primary production stimulated by nitrogen loads rather than catchment organic carbon flux was responsible for the depletion of bottom water DO in summer. Although a significant amount of phosphorus was stored in the

5 sediment, it would only be released under low bottom water DO conditions when a large quantity of POC settled in the sediment which is ultimately driven by nitrogen loading. In addition, the residence time of flood-introduced DIN can be largely influenced by a number of factors including the availability of phosphorus in flood water. It was found that DIN contributed by flood can be converted to PON by photosynthesis quickly enough to prevent being flushed out of the lakes. The current study demonstrated that it is important to reduce both TN and TP in hypoxia mitigation in estuarine systems.

### Acknowledgments

This work was supported by the Australian research council grant LP140100087 to PLMC, and the Victorian Department of Environment, Land, Water and Planning. The authors also thank DHI for provision of licenses for MIKE21 SW, MIKE3 FM

15 and ECOLAB.



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

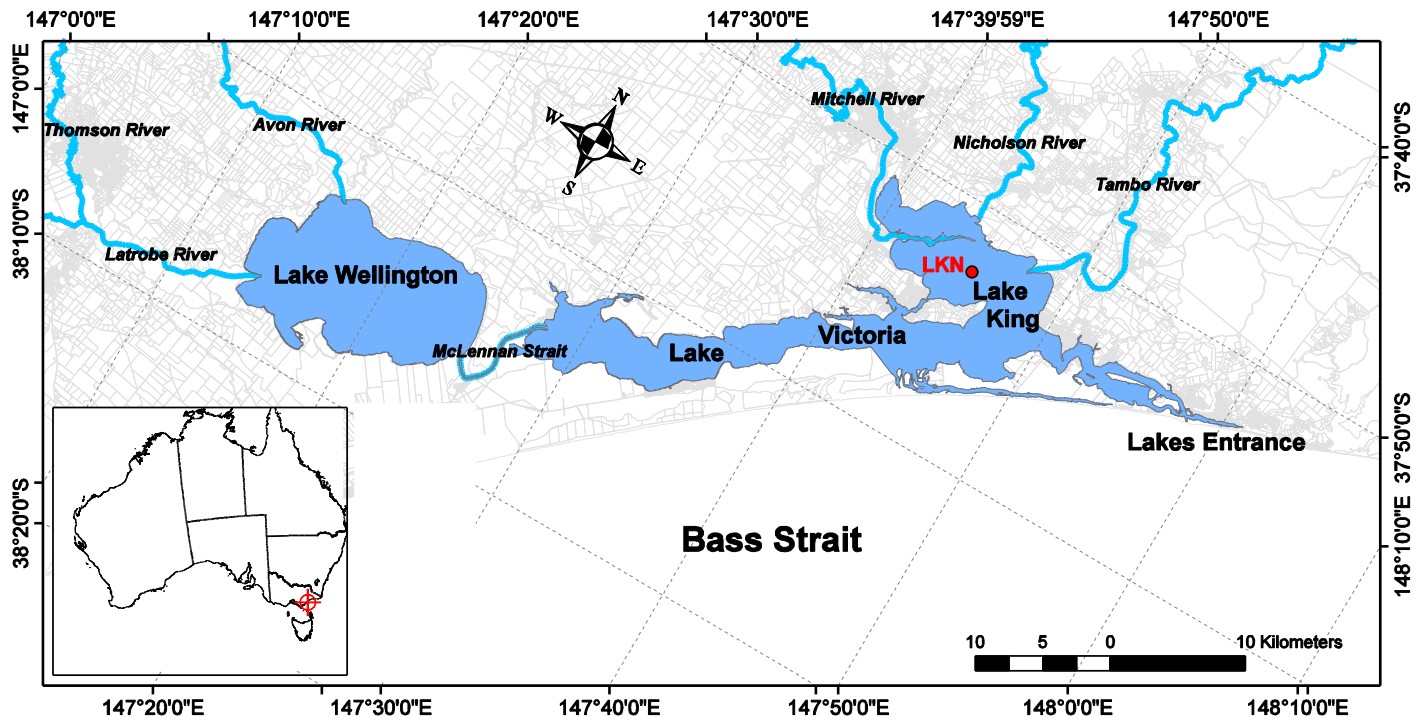

**Figure 1 Gippsland Lakes, major tributaries and the location of LKN**





**Table 1 Nutrient loads between May 2010 and July 2012**

|  | **Eastern Rivers** | **Western Rivers** |
|---|---|---|
| **DIN (tN)** | 411.92 | 1,403.76 |
| **PON* (tN)** | 681.58 | 1,197.09 |
| **DIP (tP)** | 18.45 | 77.77 |
| **PP*# (tP)** | 89.82 | 171.18 |
| **POC (tC)** | 6,815.82 | 11,970.9 |

*the refractory faction has been excluded
# PP is particulate P which is TP – DIP

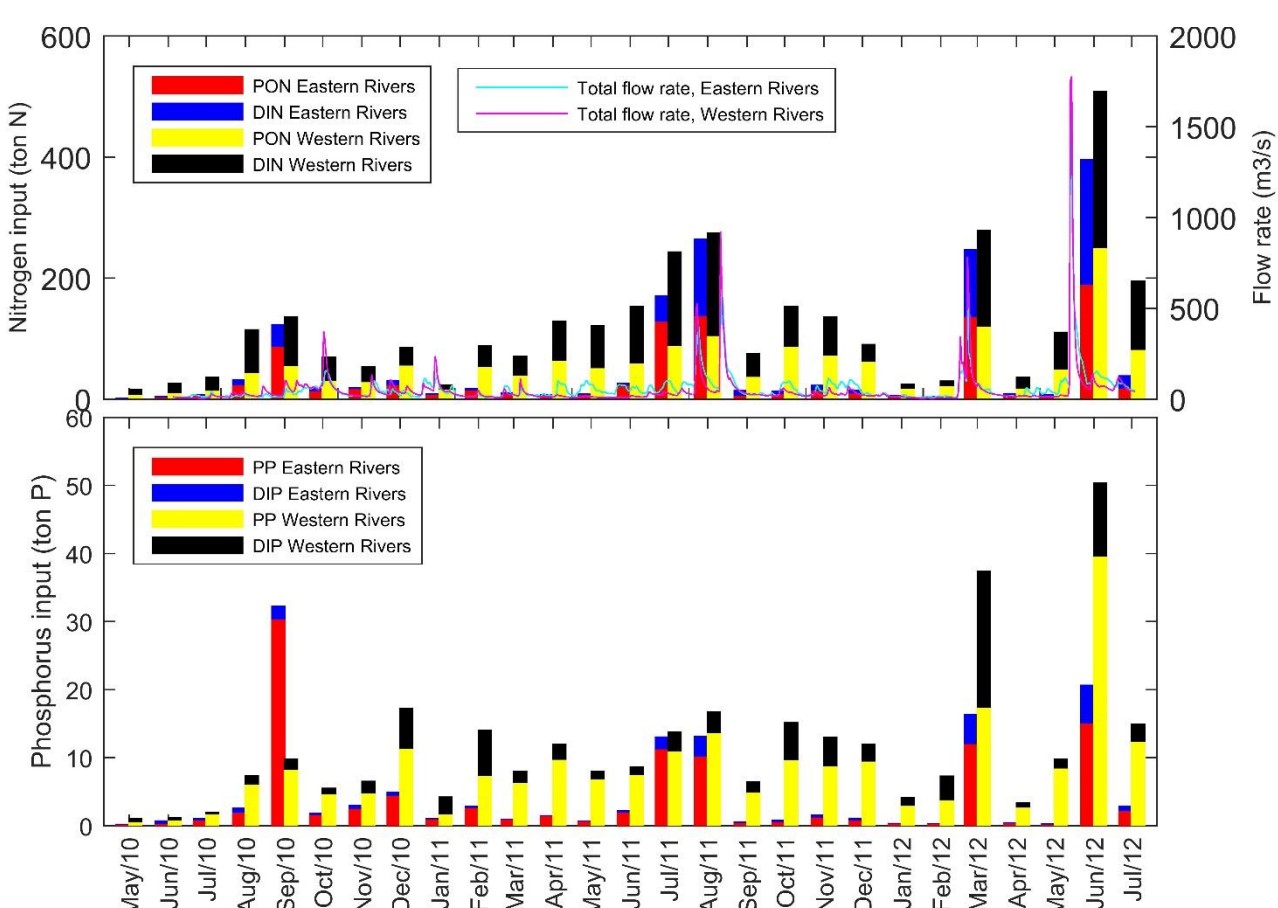

**Figure 2 Monthly catchment nutrient input and river flows**





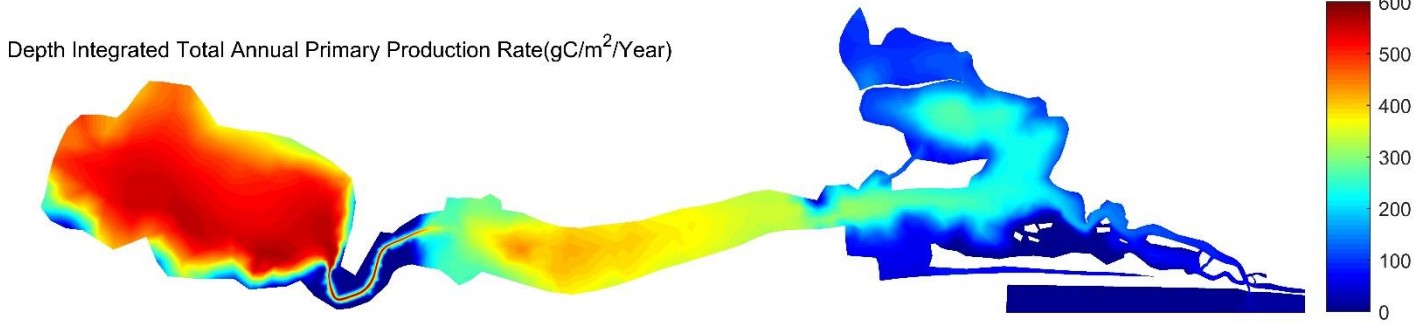

**Figure 3  Modelled depth integrated total annual primary production for the base case.**

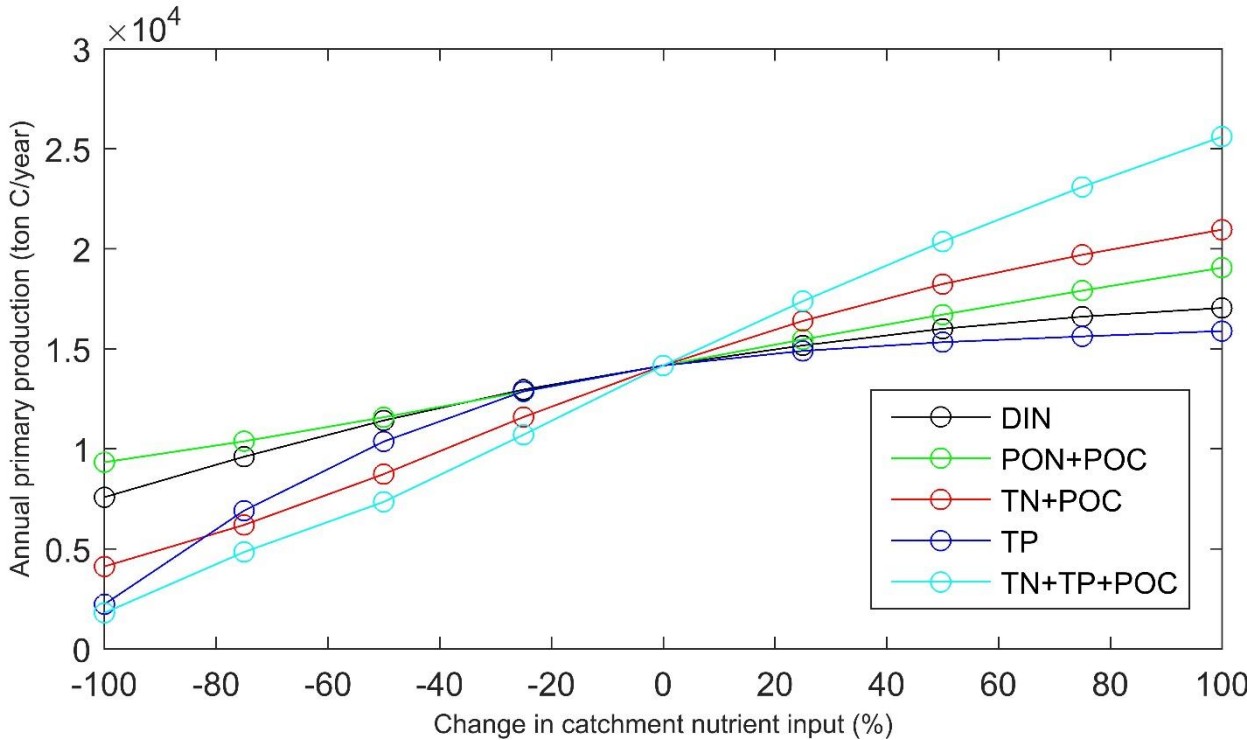

**Figure 4  Modelled total primary production in Lake King between May 2010 and July 2012: DIN, change in dissolved inorganic nitrogen load; PON+POC, change in particulate organic nitrogen and carbon loads; TN+POC, change in total nitrogen and particulate organic carbon loads; TP, change in total phosphorus load; TN+TP+POC, change in total nitrogen, total phosphorus and particulate organic carbon loads**





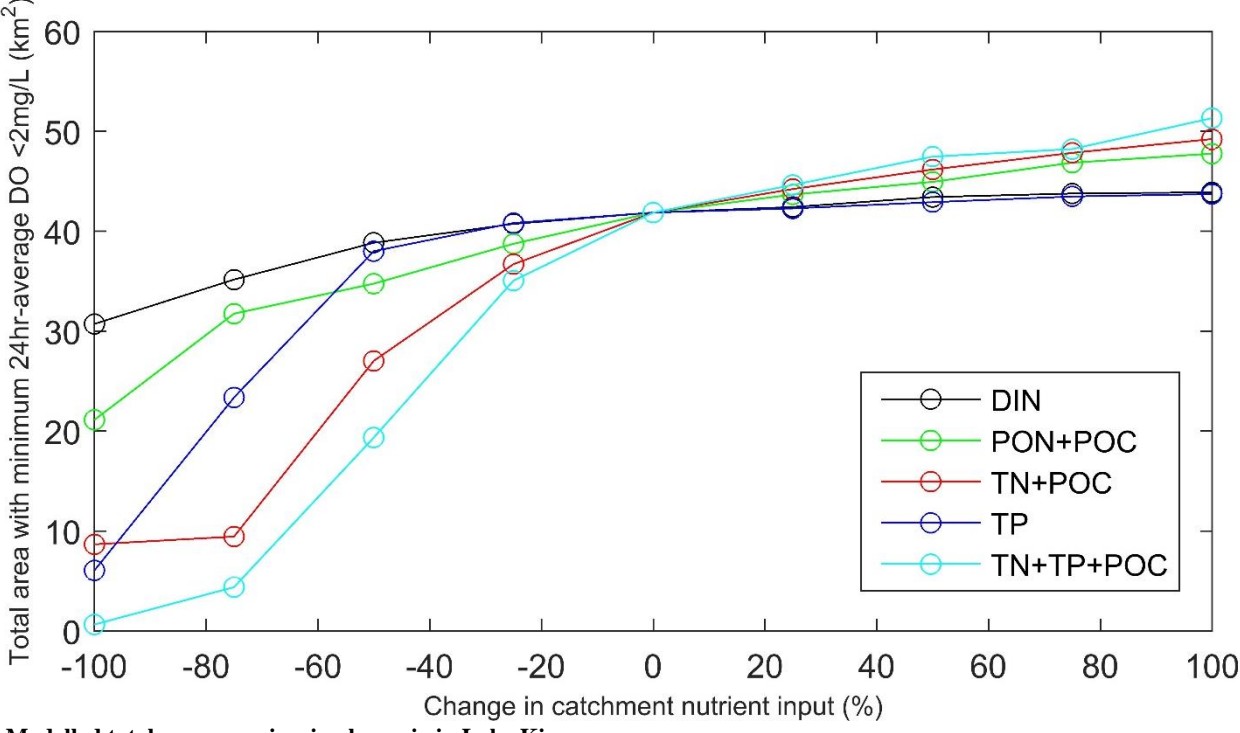

**Figure 5  Modelled total area experiencing hypoxia in Lake King**

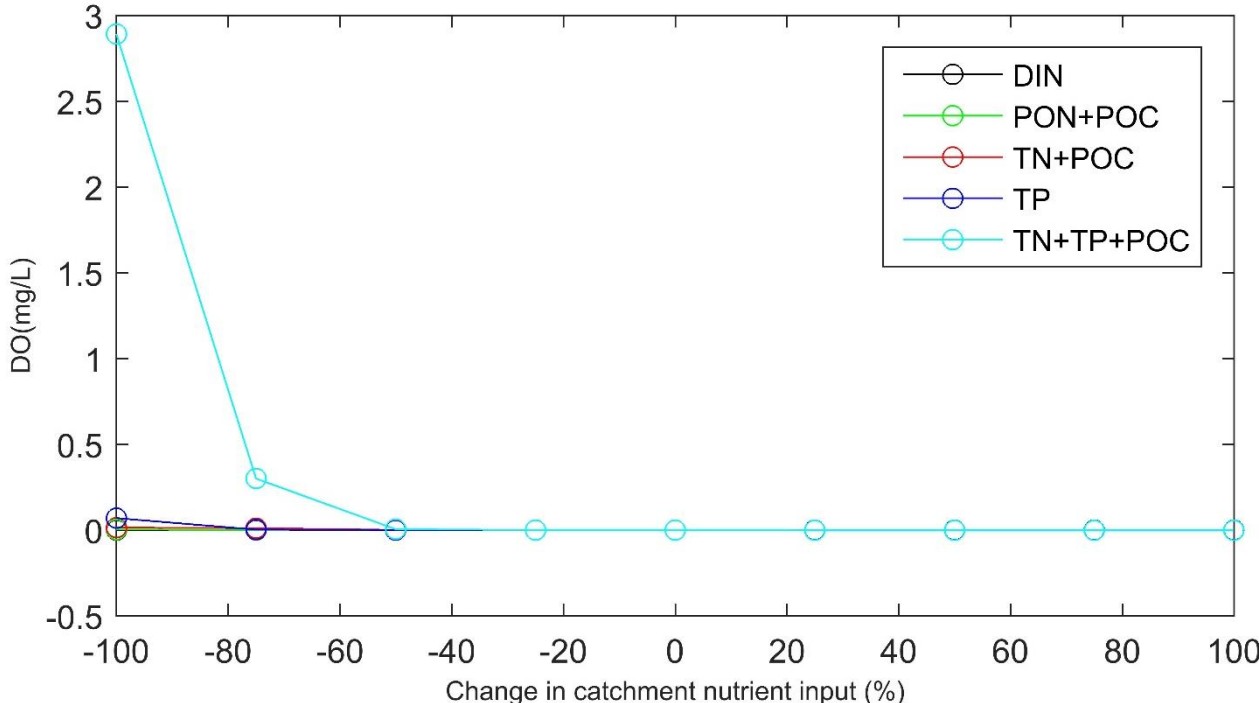

**Figure 6  Minimum bottom DO concentration at LKN**





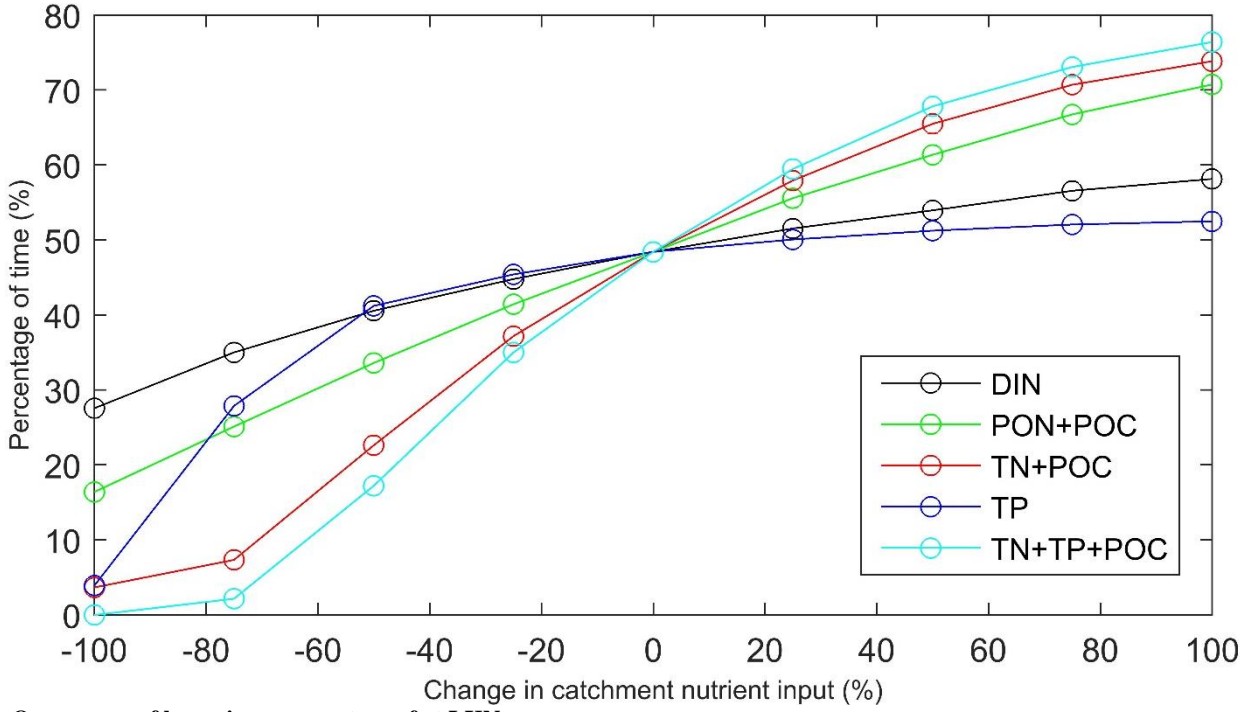

**Figure 7  Occurrence of hypoxia as percentage of at LKN**

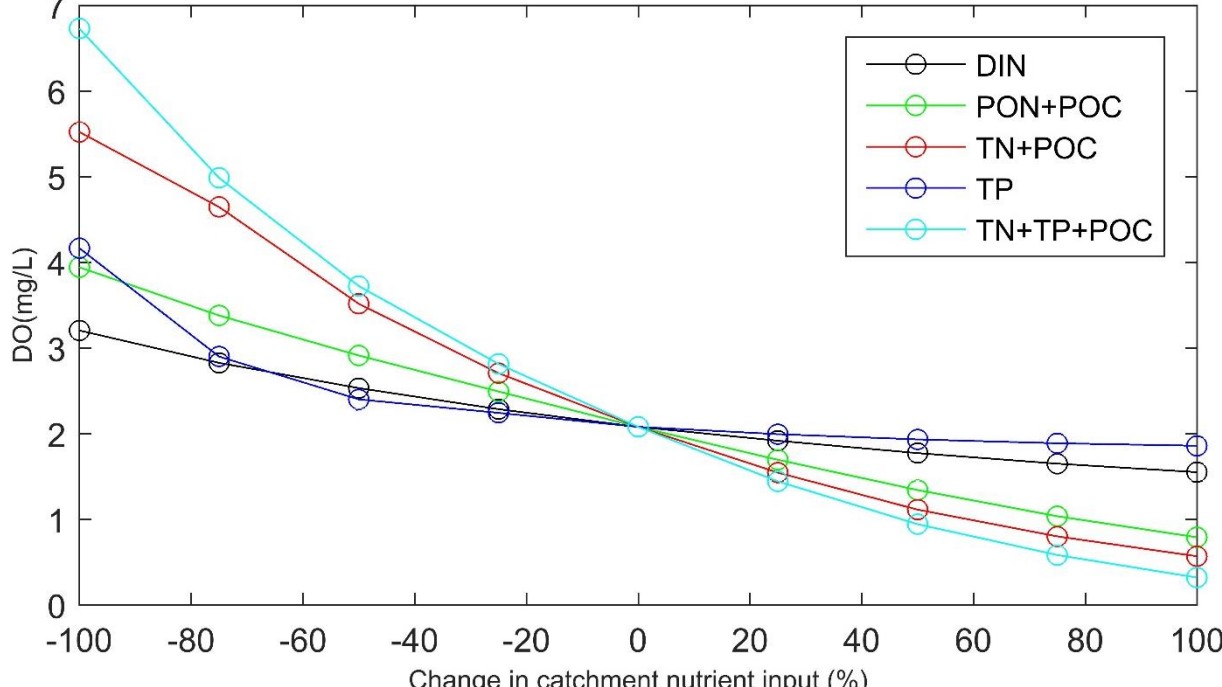

**Figure 8  Median bottom DO concentration and occurrence of hypoxia as percentage of at LKN**



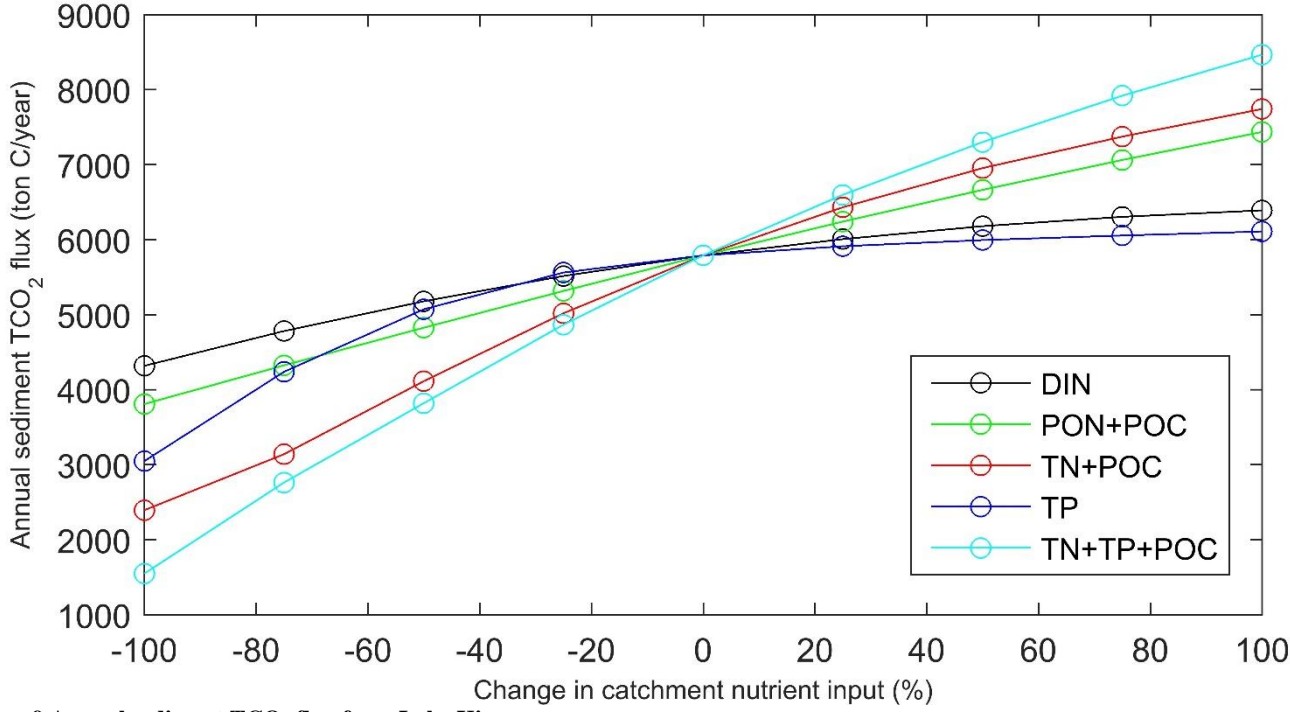

**Figure 9 Annual sediment TCO₂ flux from Lake King**

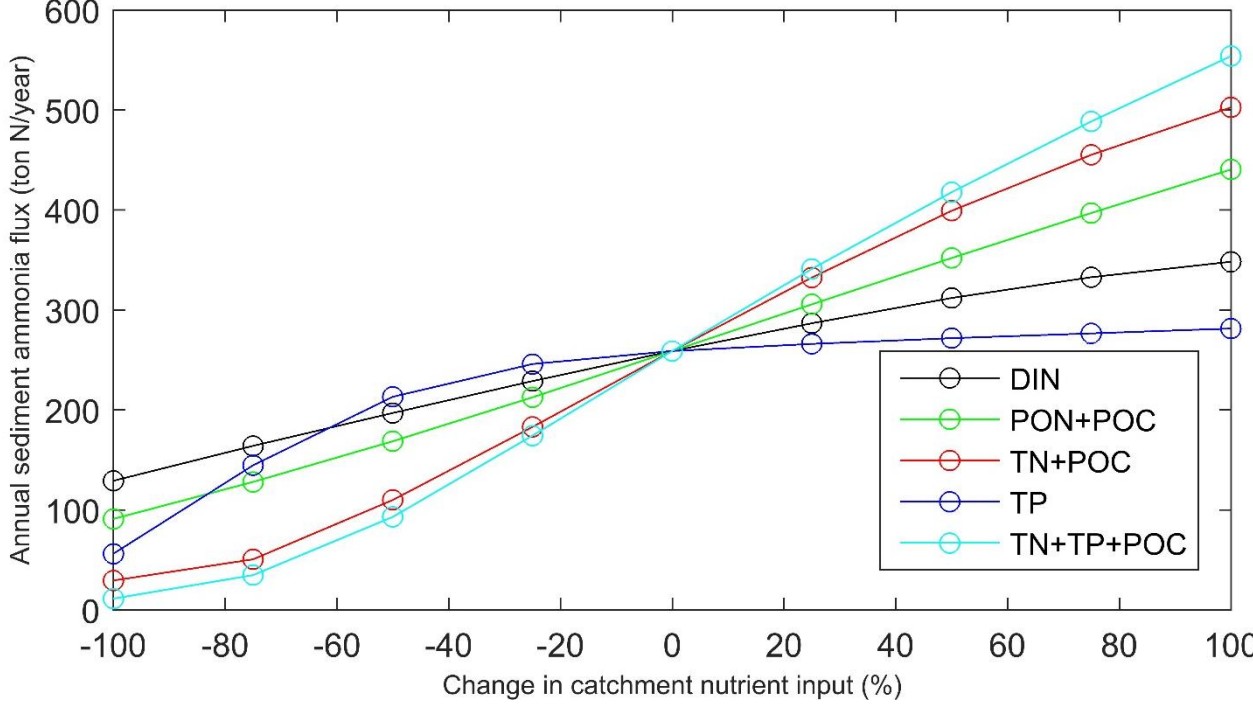

**Figure 10 Annual sediment ammonia flux from Lake King**



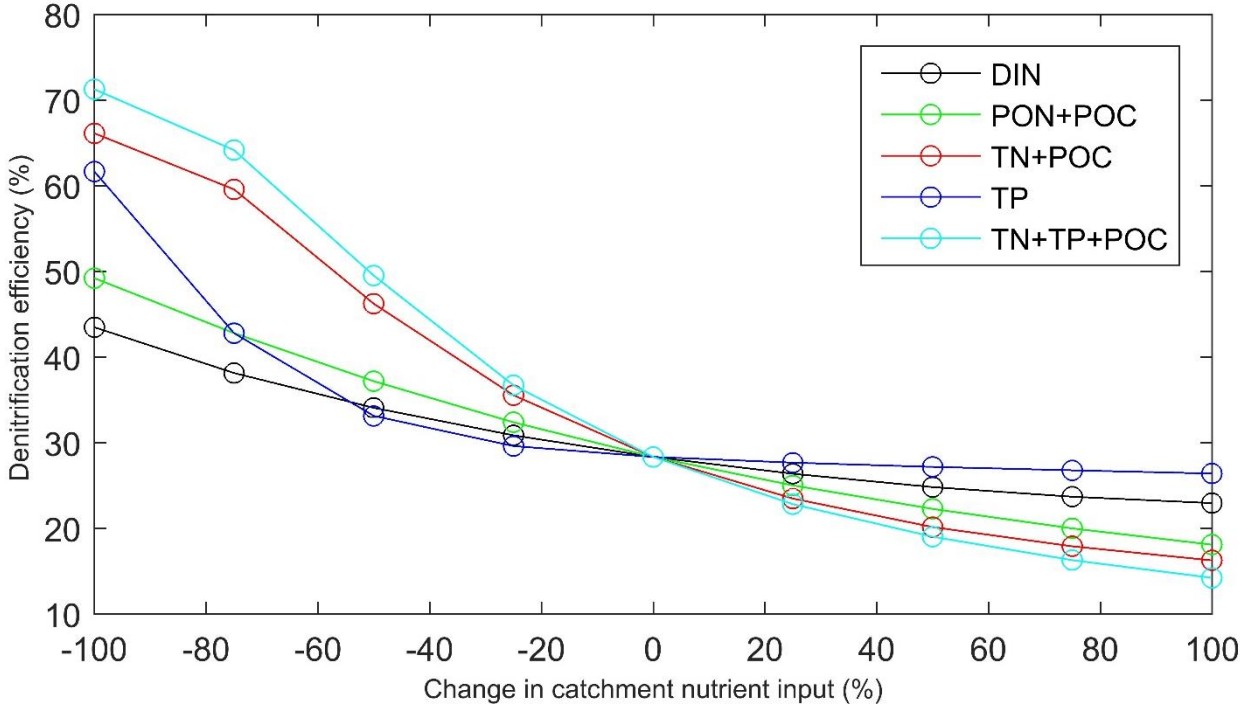

**Figure 11 Annual average denitrification efficiency from Lake King**

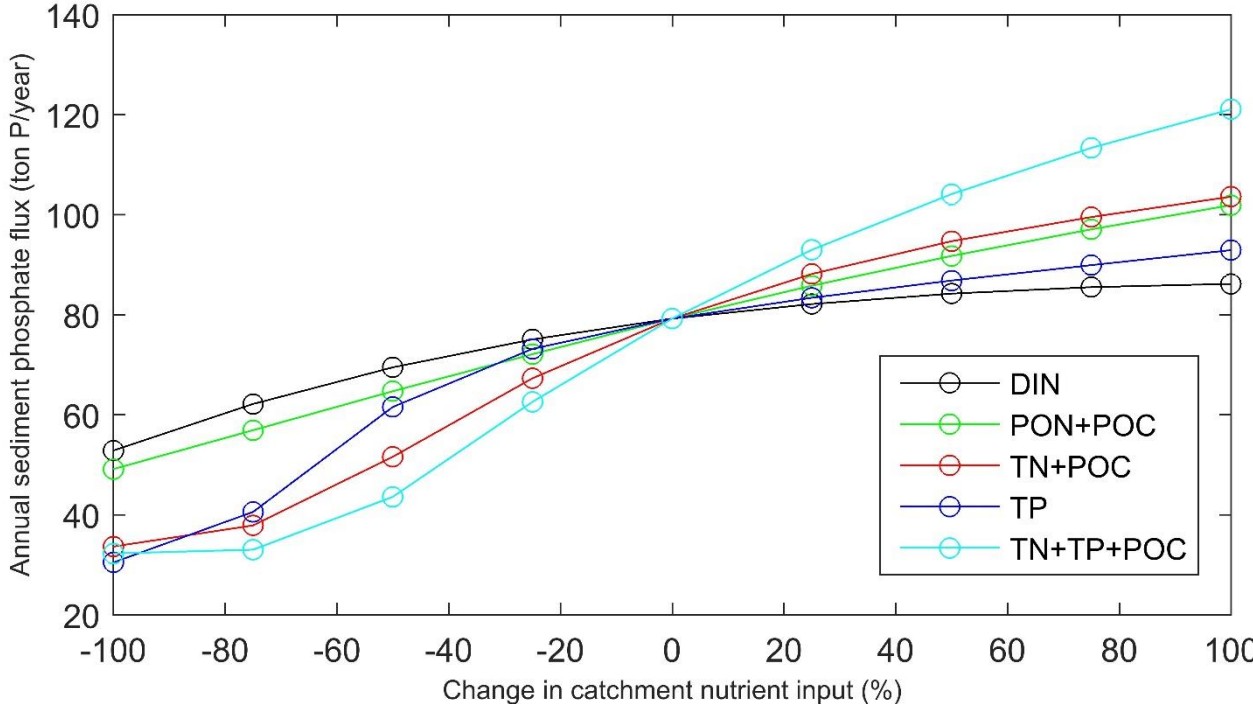

Figure 12 Annual sediment phosphate flux from Lake King

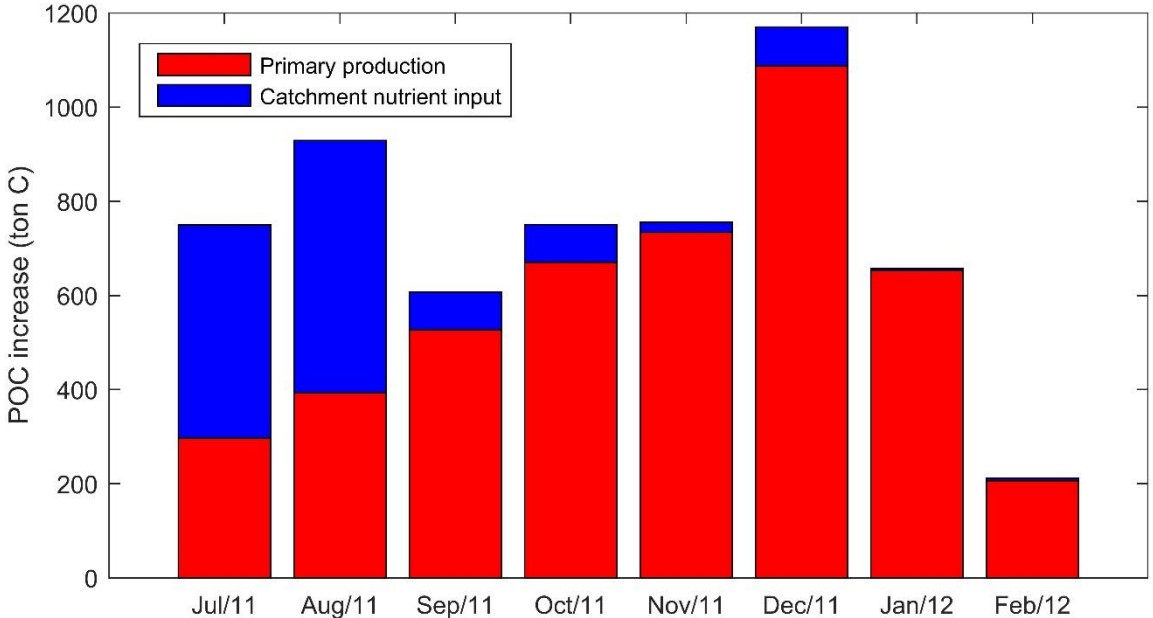

**Figure 13 Sediment POC deposition at Lake King contributed from catchment input and primary production**