# Peer review of "Effects of changes in nutrient loading and composition on hypoxia dynamics and internal nutrient cycling of a stratified coastal lagoon"

_Biogeosciences, 2017_

## Referee Comment (RC1) · Anonymous Referee #1 · 17 May 2017

The present study use a 3D hydrodynamic-biogeochemical model implemented and developed for the Gibbs lakes (previously published) to investigate different nutrient catchment scenarios and assess the relative contribution of nitrogen VS phosphorus and inorganic VS particulate catchment loads in controlling the onset and development of hypoxia in this region. The model complexity is adequate to address this issue (eg. include benthic component, resuspension and bioirrigation dynamics) and the choice of the scenario is sound.

The highlighted importance to consider the composition of catchment loads, in particular to characterize the bioavailabiliy of particulate compounds, and the described mechanisms showing the intertwined influence of both P and N inputs on hypoxia

makes this study an important contribution to the management of eutrophication in coastal areas. The abstract is succinct and reflects well the main findings of the study. The introduction is appropriated, but some conclusions of previous work should be more detailled. The methods (model description) requires a bit more details on key issues affecting the discussion (eg. benthic aspects), and on the scenario implementation (initial conditions, period considred for statistics). The readability of results section might be improved. Some section of the discussion should expand slightly on the different element addressed, ie some points are addressed very specifically without having been introduced before ( eg. P10 L 23-25).

I recommend the publication of this work after consideration of those comments and the remarks below, which I believe can be met with minor modifications.

MAJOR COMMENTS

* Sect 2 Could you clarify if "balanced" Initial Conditions for the sediment (ie, as produced following the methodology described in p 4 L 19-24) were derived for each scenario, or if all scenario starts with the same IC. In the latter case (same sed ICs for all scenario), there would be a transient period during which sediments supports the nutrient delivery, and I wonder about any temporal trends in the indicated results (ie, would the same response to catchment scenario be obtained if rates were computed over different period of the simulations). Could the author comment on this aspect and provide a justification for the period selected as a basis for scenario comparison (ie; the numbers provided in Figs 4-12) In the former case (different ICs, specifically balanced for the different catchment scenarios), I don't understand the content of P7 L 6-8.

* Sect 2.2 It would be good to have a few lines on the functionning of benthic-pelagic coupling in the model.

* P5L9 I don't understand the justification given for the estimation of the labile fraction of particulate organic input. What relates the 60% evaluated between the C/N ratios for labile OM and catchment OM, and the 60% deduced for the ratio between labile

and refractory component. Wouldn't there be a need to assign some C/N value to the refractory component to close this computation?

* P5 Last paragraph: Please provide in the text the period over which statistics presented in Figs 4-12.

* P9 L27 : It is very difficult to understand the development given here without a few lines in the model description of how the sediment module works. I think it would be much easier to decribe this mass balance with a few equations. A few points : * Zhu et al 2016 mention burial: How is burial considered in the present mass balance ? * The fact that TCO2 fluxes in the zero catchment scenario quantify the contribution from the refractory sediment stocks only is again related to the IC question above, please clarify. * L31: Why a different period is considered here (July2011 -> Jan2012). This also relates to the first question. Also on Fig. 5 the tot PP for the no-catchment scenario seems to be around 15% of the base case, and not 0.38% ? Could you explain?

MINOR COMMENTS * P2 L5 : How does hypoxia or anoxia enhance the recycling of N ?

* P2 L25 : " have been studied .. " -> could you briefly present the main conclusions of those previous study on the contribution of allocthonous/autochtonous organic matter to coastal hypoxia ?

* P2 L10 space after "."

* P2L02 lowercase "N"itrogen

* P4L27 provide the reference for validation again.

* P4L31 knowns->known

* P5 l20 space before "."

* Sect 3.2 : Please provide an explicit definition of "hypoxic area". For instance P6L15 "area covered", means area where hypoxia prevails for more than 24h? ( deduced from

axis label of Fig 5)

* P6 L18 : Why is DON mentioned here. Sect. 2.2 precise DON and DOC are not represented in the model?

* P7 L9-10 : "The ratio ..41)" : I don't understand this sentence. What do you mean. This ratio was 33% instead of 8.5 % at Lake King. What is the $R^2$ referring to ? Please rephrase

* P7 L21-22 . In the case were all scenario starts from the same ICs (see main question 1), could it also be due to an ongoing mineralization of refractory sediment stocks ?

P8 L5: Those "mechanisms" were not mentioned in the results, nor are they clearly described in the following. Clarify or remove this sentence.

P8L6 " the model simulated the transport " -> "we used the model to simulate the transport .. "

P8 "land use" -> could you expand a bit the discussion here ?

P8 L15 : were all biogeochemical processes disabled to estimate transport or only plankton uptake ?

P8 L16: Please precise how is the 70% computed and to what the $R^2$ refer.

P8 L15-17 : Please rephrase. The reader can understand the message with the next sentence but it is not clear in the present form

P8L18 : please provide explicitly the definition for TN export rate.

P9 L 16 and following: Might be rephrased for clarity. For instance using the autochtonous/allochtonous nomenclature.

P 10 L8 .. contribute "by" less than 7% "to" the ..

P10 L9 bottom water "Oxygen" depletion

P10 L23-25, please clarify or better integrate in the current discussion.

Fig 5 -> reallocate the definition given in the axis label ( min 24h ..)  to the caption or the text (or both)

Fig 6 is maybe not essential, and could be described in words.

Fig8 caption mentions again " and occurence of hypoxia .. of". Is that a Typo ?

---

## Referee Comment (RC2) · Anonymous Referee #2 · 23 Jun 2017

This is a useful paper that manages to describe a large range of nutrient load scenarios in a limited amount of space. I think the use of the model to examine feedbacks within the system and quantify the relative influence of external and internal processes, the inherent linkages between "external" and "internal" processes, and the spatial nature of these processes is the exact type of approach one should use these tools for. I did find several sections of the discussion to be confusing or misleading, and I think there are some places within the methods that need more description.

Specific comments (1) Page 3, Line 29 – the sentence that begins here and describes the turbulence closure schemes is organized and worded in an odd way – please rewrite it.

(2) A model of bio-irrigation is mentioned (Page 4, Line 15) without any real description of it. I realize there is another paper describing this model, and after reading that paper (Zhu et al. 2016) I am convinced that this bio-irrigation model could be described in the current paper in just a few sentences to provide the basic details.

(3) Page 4, Lines 3-5 – has the wave model been validated at all? It is not in this paper, and from what I can tell, was not validated in Zhu et al. (2016). Either some validation is due in this paper, or the authors should cite where the validation exists.

(4) Page 4, Lines 23-24: A" spatially-varying sediment iron-bound phosphate…." Is mentioned as an initial condition, but it is not clear at all how iron-phosphorus dynamics are represented by the model, and these dynamics are not really described in Zhu et al. (2016). I assume you are not modeling iron explicitly, but rather that there is an adsorption between iron-bound and free phosphate that is modulated by oxygen and assumes unlimited adsorption (Zhu et al. 2016 include an adsorption parameter in the appendix table). You could clarify this with two additional sentences I think. For this comment and comment #2, I believe strongly that a given paper should attempt to be a stand-alone document, and cannot completely rely on a previous paper to describe the model. Obviously you cannot re-write the entire model description, but each model component that is highlighted should have a basic description of it and the original source of the details.

(5) There are a fair amount of basic grammatical errors in the text – please read over carefully to correct this.

(6) Page 7, Line 9: I assume that TPP is gross photosynthesis. If so, while it is relevant to relate TPP to the external POC loading, I think it is less useful for comparing TPP to sediment CO2 fluxes. I think it would be more appropriate to relate TPP-R (net phytoplankton production) to sediment CO2 fluxes, because it is the net production that potentially yields carbon that can sink to sediments to support CO2 production.

(7) Page 7, Line 17: Please define explicitly how you computed denitrification efficiency.

(8) Page 9, Lines 6-8: Please provide values for bottom shear stress to help the reader understand what "very low" means, relative to the rest of Gippsland lakes and other systems.

(9) Page 10, Line 23-24: I think it is worth stating clearly what the mechanism is that limits internal phosphorus loading with elevated nitrate. Although you cite literature, the mechanism is not intuitive and perhaps not widely known.

(10) Page 11, Lines 4-5: What do you mean by the sentence "However, initial input of catchment phosphorus......"? What analysis or model run is this based on? There are not analyses in the paper to support this, and in the absence of such a statement, this sentence appears to contradict the one that came just before it

(11) On Page 10, Lines 9-10, you state that bottom oxygen depletion in Lake King is primarily related to nutrient inputs and phytoplankton production, and your scenarios indicate that elevated TP loads did little to stimulate additional TPP and hypoxia. But then on Page 11, Lines 12-15, you indicate that there could be a recalcitrance of the system in the face of modest nutrient reductions due to internal phosphorus loading from sediment stores, which seems to contradict the prior statements. Please clarify the specifics of this in the manuscript if that is possible, as it leaves the reader wanting for a resolution.

(12) You also indicate a 5-10 year time frame for the exhaustion of internal P stores, but what is that based upon? It would seem easy to cut off new nutrient inputs and re-run the model for 5-10 more years of no-new nutrient loads and quantify for how many years the sediments continue to release phosphorus without new watershed inputs.

(13) The paragraph ending Page 11 needs grammatical editing.

(14) Conclusion section, Page 12, lines 1-3: The first two statements of this paragraph, again, appear to contradict one another. The first sentence says hypoxia is driven

by stratification and sediment carbon enrichment, while the second says nitrogen-stimulated primary production was responsible for DO depletion. So which is it? Again, why would the internal phosphorus loading matter if nitrogen is the key limiting nutrient? You did show that TP loading increases stimulated TPP beyond what TN and POC stimulated – so perhaps some improved wording would help.

(15) Figure 2: I think it would be easier to see the flow record if it had its own panel

(16) Units: I understand the value of using tons to represent large numbers, but it might also help to indicate, perhaps in the text, what the sediment-water ammonium and phosphate fluxes were in commonly used units (micromole/m-2/h-1). Perhaps simply contrasting the rates at the highest nutrient increase and larges reduction. This would help compare these numbers with other systems.

---

## Author Comment (AC1) · 28 Jun 2017

We would like to thank reviewer 1 for his/her thorough reading of our manuscript and for the thoughtful comments and constructive feedback which we believe has helped us to improve the quality of this manuscript. We have carefully reviewed the comments and have prepared the following plan to address them in the revised version:

1) Sect 2 Could you clarify if "balanced" Initial Conditions for the sediment (ie, as produced following the methodology described in p 4 L 19-24) were derived for each scenario, or if all scenario starts with the same IC. In the latter case (same sed ICs for all scenario), there would be a transient period during which sediments supports the

nutrient delivery, and I wonder about any temporal trends in the indicated results (ie, would the same response to catchment scenario be obtained if rates were computed over different period of the simulations). Could the author comment on this aspect and provide a justification for the period selected as a basis for scenario comparison (ie; the numbers provided in Figs 4-12) In the former case (different ICs, specifically balanced for the different catchment scenarios), I don't understand the content of P7 L 6-8.

We will add a couple of sentences to make it clear that all the scenario had the same initial condition which was derived for the base case. The scenario with no catchment nutrient loading showed that the sediment N (both organic and inorganic) depleted within a few month from the start of the simulation. This short transition period did not significantly affect the overall results. However, P may have a much longer residence time in the system. There are a few reasons for why we have chosen the selected period for scenario comparison: 1) a good set of monitoring data was collected during that period; 2) the span of the simulation covered both dry and wet periods; 3) and, we have a good knowledge of the sediment geochemistry over the same period.

2) Sect 2.2 It would be good to have a few lines on the functioning of benthic-pelagic coupling in the model.

We will add a few sentences to address this.

3) P5L9 I don't understand the justification given for the estimation of the labile fraction of particulate organic input. What relates the 60% evaluated between the C/N ratios for labile OM and catchment OM, and the 60% deduced for the ratio between labile and refractory component. Wouldn't there be a need to assign some C/N value to the refractory component to close this computation?

The model had organic matter in the labile and refractory portions. Because there was no C data from the catchment load, we estimate a C:N weight ratio of 10 for catchment C loads based on the recorded N loads. The C:N ratio can be a good indicator of the lability of organic matter. If we assume Redfield (C:N=5.7) material is

labile then 5.7/10∼=0.6 can be a good estimation of the labile portion for the catchment organic matter. The estimate was very close to the ones published for some other major rivers around the world (P5L21). The model does not close the computation based on C/N ratio but there was conversion from labile to refectory which is associated with mineralisation.

4) P5 Last paragraph: Please provide in the text the period over which statistics presented in Figs 4-12.

To be added.

5) P9 L27 : It is very difficult to understand the development given here without a few lines in the model description of how the sediment module works. I think it would be much easier to decribe this mass balance with a few equations. A few points : * Zhu et al 2016 mention burial: How is burial considered in the present mass balance ? * The fact that TCO2 fluxes in the zero catchment scenario quantify the contribution from the refractory sediment stocks only is again related to the IC question above, please clarify. * L31: Why a different period is considered here (July2011 -> Jan2012). This also relates to the first question. Also on Fig. 5 the tot PP for the no-catchment scenario seems to be around 15% of the base case, and not 0.38% ? Could you explain?

There is an equation (P9L33) that explains how the mass balance was worked out but we will add more descriptions to make it easier to understand. For the mass balance calculated here was not affected by burial. As organic matters will only be removed from the model by burial when the total depth exceeds a depth threshold (0.2m in this case) Since each scenario had the same initial condition, the TCO2 Fluxes in the zero catchment scenario can be a good estimate for what was contributed by refectory sediment C. We will make it clear that the mass balance was carried out for July 2011 to Jan 2012. The purpose of the mass balance was to compare the importance of catchment carbon and primary production to the development of hypoxia. The reason we chose this period was because during this period 1) large quantity of catchment

organic carbon was introduced by a flood 2) and hypoxia was developed in this period. Figure 5 present the results over the entire 2-year simulation. The percentage (either 0.38% or 15%) presented here were all relative. For example, let's assume the PP is 15 ton/month over 12 month, and the total catchment carbon was 100 ton/year but 80 ton was introduced within one wet month. Then the ratio of PP: Catchment C would be 12/100 for the 12-month period but 1/80 for the wet month.

6) MINOR COMMENTS * P2 L5 : How does hypoxia or anoxia enhance the recycling of N ?

We suppose hypoxia/anoxia would reduce the production of NO3 through nitrification and thus affect the denitrification process.

7) P2 L25 : " have been studied .. " -> could you briefly present the main conclusions of those previous study on the contribution of allocthonous/autochtonous organic matter to coastal hypoxia ?

To be updated.

8) P2 L10 space after "."

To be updated.

9) P2L02 lowercase "N"itrogen

To be updated.

10) P4L27 provide the reference for validation again.

To be updated.

11) P4L31 knowns->known

To be updated.

12) P5 l20 space before "."

To be updated.

13) Sect 3.2 : Please provide an explicit definition of "hypoxic area". For instance P6L15 "area covered", means area where hypoxia prevails for more than 24h? ( deduced from axis label of Fig 5)

To be updated.

14) P6 L18 : Why is DON mentioned here. Sect. 2.2 precise DON and DOC are not represented in the model?

It should be PON. To be updated.

15) P7 L9-10 : "The ratio ..41)" : I don't understand this sentence. What do you mean. This ratio was 33% instead of 8.5 % at Lake King. What is the R2 referring to ? Please rephrase

To be updated. This is the trend line when plot TPP vs TCO2.

16) P7 L21-22 . In the case were all scenario starts from the same ICs (see main question 1), could it also be due to an ongoing mineralization of refractory sediment stocks ?

Strictly speaking, yes, the ongoing mineralization of refractory sediment stocks contribute majority of the SOD for the zero catchment scenario. This SOD was small but relatively constant over time. However, it was really stratification that prevented oxygen replenishment and induced hypoxia.

17) P8 L5: Those "mechanisms" were not mentioned in the results, nor are they clearly described in the following. Clarify or remove this sentence.

To be updated.

18) P8L6 " the model simulated the transport " -> "we used the model to simulate the transport .. "

To be updated.

19) P8 "land use" -> could you expand a bit the discussion here ?

To be updated.

20) P8 L15 : were all biogeochemical processes disabled to estimate transport or only plankton uptake ?

All biogeochemical processes including plankton uptake was disabled.

21) P8 L16: Please precise how is the 70% computed and to what the R2 refer.

This was calculated by plot TN remaining vs TN input, 70% is the trend line with a slope of 0.7.

22) P8 L15-17 : Please rephrase. The reader can understand the message with the next sentence but it is not clear in the present form

To be updated.

23) P8L18 : please provide explicitly the definition for TN export rate.

To be updated.

24) P9 L 16 and following: Might be rephrased for clarity. For instance using the autochtonous/allochtonous nomenclature.

To be updated.

25) P 10 L8 .. contribute "by" less than 7% "to" the .. P10 L9 bottom water "Oxygen" depletion

To be updated.

26) P10 L23-25, please clarify or better integrate in the current discussion.

To be updated.

27) Fig 5 -> reallocate the definition given in the axis label ( min 24h ..) to the caption or the text (or both)

To be updated.

28) Fig 6 is maybe not essential, and could be described in words.

Figure 6 will be deleted

29) Fig8 caption mentions again " and occurence of hypoxia .. of". Is that a Typo ?

To be updated.

---

## Author Comment (AC2) · 28 Jun 2017

We thank reviewer 2 for his/her thoroughly reading of our manuscript and for the thoughtful comments and constructive feedback which we believe has helped us to improve the quality of this manuscript. We have carefully reviewed the comments and have revised the manuscript accordingly. Our response are given in point-by-point manner after your comments:

1) Page 3, Line 29 – the sentence that begins here and describes the turbulence closure schemes is organized and worded in an odd way – please re-

[Figure]

To be updated.

2) A model of bio-irrigation is mentioned (Page 4, Line 15) without any real description of it. I realize there is another paper describing this model, and after reading that paper (Zhu et al. 2016) I am convinced that this bio-irrigation model could be described in the current paper in just a few sentences to provide the basic details.

To be updated. We will improve the model description to make it more readable.

3) Page 4, Lines 3-5 – has the wave model been validated at all? It is not in this paper, and from what I can tell, was not validated in Zhu et al. (2016). Either some validation is due in this paper, or the authors should cite where the validation exists.

To be updated.

4) Page 4, Lines 23-24: A" spatially-varying sediment iron-bound phosphate. . .." Is mentioned as an initial condition, but it is not clear at all how iron-phosphorus dynamics are represented by the model, and these dynamics are not really described in Zhu et al. (2016). I assume you are not modeling iron explicitly, but rather that there is an adsorption between iron-bound and free phosphate that is modulated by oxygen and assumes unlimited adsorption (Zhu et al. 2016 include an adsorption parameter in the appendix table). You could clarify this with two additional sentences I think. For this comment and comment #2, I believe strongly that a given paper should attempt to be a stand-alone document, and cannot completely rely on a previous paper to describe the model. Obviously you cannot re-write the entire model description, but each model component that is highlighted should have a basic description of it and the original source of the details.

Yes, iron was not modelled explicitly but treated as a spatially varying constant estimated by previous studies. We will improve the model description to make it more readable.

5) There are a fair amount of basic grammatical errors in the text – please read over

carefully to correct this.

We will conduct a thorough proofreading.

6) Page 7, Line 9: I assume that TPP is gross photosynthesis. If so, while it is relevant to relate TPP to the external POC loading, I think it is less useful for comparing TPP to sediment CO2 fluxes. I think it would be more appropriate to relate TPP-R (net phytoplankton production) to sediment CO2 fluxes, because it is the net production that potentially yields carbon that can sink to sediments to support CO2 production.

Yes, TPP is the gross photosynthesis. It is relatively easy to track and extract from the model. However, the net phytoplankton production is not a readily available output. On the other hand, TPP-R is strongly correlated to TPP. Therefore we consider the use of TPP in current study was also appropriate.

7) Page 7, Line 17: Please define explicitly how you computed denitrification efficiency.

To be updated.

8) Page 9, Lines 6-8: Please provide values for bottom shear stress to help the reader understand what "very low" means, relative to the rest of Gippsland lakes and other systems.

To be updated.

9) Page 10, Line 23-24: I think it is worth stating clearly what the mechanism is that limits internal phosphorus loading with elevated nitrate. Although you cite literature, the mechanism is not intuitive and perhaps not widely known.

In line 25, we stated that: the model showed that the increase in oxidised depth of the sediment was limited due to increased sediment oxygen demand and low diffusion rate of nitrate in the sediment.

10) Page 11, Lines 4-5: What do you mean by the sentence "However, initial input of catchment phosphorus. . .. . ."? What analysis or model run is this based on? There

are not analyses in the paper to support this, and in the absence of such a statement, this sentence appears to contradict the one that came just before it

We came to this conclusion by comparing the zero TP scenario to the other ones. If there is no TP from the catchment, there would be no primary production. There must be P input for primary production to trap the N, especially IN. This P would likely be contributed from the catchment, because the sediment P release would not occur until bottom oxygen was depleted.

11) On Page 10, Lines 9-10, you state that bottom oxygen depletion in Lake King is primarily related to nutrient inputs and phytoplankton production, and your scenarios indicate that elevated TP loads did little to stimulate additional TPP and hypoxia. But then on Page 11, Lines 12-15, you indicate that there could be a recalcitrance of the system in the face of modest nutrient reductions due to internal phosphorus loading from sediment stores, which seems to contradict the prior statements. Please clarify the specifics of this in the manuscript if that is possible, as it leaves the reader wanting for a resolution.

This was because sediment P rather than the catchment P supplied majority of the P to support TPP and hypoxia. The sediment P originated from the catchment. Therefore, reduction in catchment P will eventually reduce the P accumulation in the sediment. We will clarify this in the manuscript.

12) You also indicate a 5-10 year time frame for the exhaustion of internal P stores, but what is that based upon? It would seem easy to cut off new nutrient inputs and re-run the model for 5-10 more years of no-new nutrient loads and quantify for how many years the sediments continue to release phosphorus without new watershed inputs

This statement was based on the long-term studies of some European lakes (Schindler and Hecky, 2009), the 5-10 year was an expectation only. We will remove this statement to avoid confusion.

13) The paragraph ending Page 11 needs grammatical editing.

To be updated.

14) Conclusion section, Page 12, lines 1-3: The first two statements of this paragraph, again, appear to contradict one another. The first sentence says hypoxia is driven by stratification and sediment carbon enrichment, while the second says nitrogen- stimulated primary production was responsible for DO depletion. So which is it? Again, why would the internal phosphorus loading matter if nitrogen is the key limiting nutrient? You did show that TP loading increases stimulated TPP beyond what TN and POC stimulated – so perhaps some improved wording would help.

To be updated. We should say that the carbon enrichment was caused by primary production. In the Gippsland Lakes, either N or P could be the limiting nutrient depending on catchment input and sediment geochemistry, although we found N was typically the limiting one. On the other hand, both field studies and model results showed that it was sediment P release that supported the summer blooms. In fact, Figure 4 showed that increases in TP loading were not as effective as the increase TN in terms of stimulating TPP.

15) Figure 2: I think it would be easier to see the flow record if it had its own panel

Figure 2 to be updated.

16) Units: I understand the value of using tons to represent large numbers, but it might also help to indicate, perhaps in the text, what the sediment-water ammonium and phosphate fluxes were in commonly used units (micromole/m-2/h-1). Perhaps simply contrasting the rates at the highest nutrient increase and larges reduction. This would help compare these numbers with other systems.

We will include a few key figures in molar unit in text to make it more comparable to other systems.

---

## Author Response (AR1)

18 July, 2017

Professor Caroline P. Slomp
Handling Associate Editor
Biogeosciences
Princetonplein 9
Room 1.08
3584 CC  UTRECHT
The Netherlands

Dear Professor Slomp,

Please find attached revision of the manuscript bg-2017-132, entitled " Effects of changes in

nutrient loading and composition on hypoxia dynamics and internal nutrient cycling of a stratified

coastal lagoon". We have carefully considered the comments from the reviewers and undertaken

the suggested changes as summarised below in this document. In addition further other editorial

improvements have been made throughout the manuscript to improve the readability and clarity

for readers of Biogeosciences. The modified version with track changes is included at end of the

response to reviewers.

We hope the manuscript now meets your expectations for publication and if you have any further

requirements we would be pleased to make any necessary changes.

Yours Sincerely

Yafei Zhu

**Response to the 1st Reviewer's comments**

*Sect 2 Could you clarify if "balanced" Initial Conditions for the sediment (ie, as produced following the methodology described in p 4 L 19-24) were derived for each scenario, or if all scenario starts with the same IC. In the latter case (same sed ICs for all scenario), there would be a transient period during which sediments supports the nutrient delivery, and I wonder about any temporal trends in the indicated results (ie, would the same response to catchment scenario be obtained if rates were computed over different period of the simulations). Could the author comment on this aspect and provide a justification for the period selected as a basis for scenario comparison (ie; the numbers provided in Figs 4-12) In the former case (different ICs, specifically balanced for the different catchment scenarios), I don't understand the content of P7 L 6-8.*

We have added a couple of sentences to make it clear that all the scenario had the same initial condition which was derived for the base case: "…To ensure the consistency of the analyses and comparisons, all scenarios had the same initial conditions including the sediment nutrient inventory which was derived from the base case…."

The scenario with no catchment nutrient loading showed that the sediment N (both organic and inorganic) depleted within a few months from the start of the simulation. This short transition period did not significantly affect the overall results. However, P may have a much longer residence time in the system. There are a few reasons for why we have chosen the selected period for scenario comparison: 1) a good set of monitoring data was collected during that period; 2) the span of the simulation covered both dry and wet periods; 3) and, we have a good knowledge of the sediment geochemistry over the same period

*Sect 2.2 It would be good to have a few lines on the functioning of benthic-pelagic coupling in the model.*

We have expanded the model description section as recommended:"… The model included three groups of phytoplankton which were N-fixing cyanobacteria, vertically-migrating dinoflagellates and fast-growing diatoms. One group of grazers was included and configured to avoid grazing on cyanobacteria. The mortality of phytoplankton together with the catchment input was the major source of organic matter. The organic matter was represented by particulate organic carbon (POC), PON and particulate organic phosphorus (POP) and was further divided into labile and refractory fractions. To simplify, the dissolved organic carbon/nutrients and the hydrolysis process (conversion from particulate to dissolved organic carbon/nutrient) have not been modelled explicitly. Instead, the model has been configured in the way that mineralisation of particulate organic carbon /nutrient took place without going through hydrolysis first. An accumulation of organic matter layer at the floor was formed due to settling. Some of the accumulated organic matter would deposit in the sediment and some would return to water column by resuspension. The rates of disposition and resuspension were calculated based on the modelled local shear stress and the critical shear stress defined for disposition and resuspension…."

***P5L9 I don't understand the justification given for the estimation of the labile fraction of particulate organic input. What relates the 60% evaluated between the C/N ratios for labile OM and catchment OM, and the 60% deduced for the ratio between labile and refractory component. Wouldn't there be a need to assign some C/N value to the refractory component to close this computation?***

The model had organic matter in the labile and refractory portions. Because there was no C data from the catchment load, we estimate a C:N weight ratio of 10 for catchment C loads based on the recorded N loads. The C:N ratio can be a good indicator of the lability of organic matter. If we assume Redfield (C:N=5.7) material is labile then 5.7/10~=0.6 can be a good estimation of the labile portion for the catchment organic matter. The estimate was very close to the ones published for some other major rivers around the world (P5L21). The model does not close the computation based on C/N ratio but there was conversion from labile to refectory which is associated with mineralisation.

***P5 Last paragraph: Please provide in the text the period over which statistics presented in Figs 4-12.***

We have specified the period for which the statistics were calculated: "… The response of bottom water DO and sediment processes to different nutrient scenarios over the two-year simulation period between May 2010 and July 2012 were analysed and discussed, with the focus on the central basins of northern Lake King where the most severe hypoxia, highest sediment DIP fluxes and cyanobacterial blooms were located."

***P9 L27 : It is very difficult to understand the development given here without a few lines in the model description of how the sediment module works. I think it would be much easier to decribe this mass balance with a few equations. A few points : * Zhu et al 2016 mention burial: How is burial considered in the present mass balance ? * The fact that TCO2 fluxes in the zero catchment scenario quantify the contribution from the refractory sediment stocks only is again related to the IC question above, please clarify. * L31: Why a different period is considered here (July2011 -> Jan2012). This also relates to the first question. Also on Fig. 5 the tot PP for the no-catchment scenario seems to be around 15% of the base case, and not 0.38% ? Could you explain?***

We have improved the model description. There was an equation (P9L33) that explained how the mass balance was worked out. The mass balance calculated here was not affected by burial. As organic matter will only be removed from the model by burial when the total depth exceeds a depth threshold (0.2m in this case).

Since each scenario had the same initial condition, the TCO2 Fluxes in the zero catchment scenario can be a good estimate for what was contributed by refectory sediment C.

We have made it more obvious that the mass balance was carried out for July 2011 to Jan 2012. The purpose of the mass balance was to compare the importance of catchment carbon and primary production to the development of hypoxia. The reason we chose this period was because during this

period 1) large quantity of catchment organic carbon was introduced by a flood 2) and hypoxia was developed in this period.

Figure 5 presents the results over the entire 2-year simulation. The percentage (either 0.38% or 15%) presented here were all relative. For example, let's assume the PP is 15 ton/month over 12 month, and the total catchment carbon was 100 ton/year but 80 ton was introduced within one wet month. Then the ratio of PP: Catchment C would be 12/100 for the 12-month period but 1/80 for the wet month.

**MINOR COMMENTS * P2 L5 : How does hypoxia or anoxia enhance the recycling of N ?**

We suppose hypoxia/anoxia would reduce the production of NO3 through nitrification and thus affect the denitrification process.

**P2 L25 : " have been studied .. " -> could you briefly present the main conclusions of those previous study on the contribution of allocthonous/autochtonous organic matter to coastal hypoxia ?**

Updated: "…Paerl et al. (1998) showed that hypoxia in estuaries can be stimulated either by internal generated or external supplied organic matters depending on the meteorological and hydrological conditions…."

**P2 L10 space after "."**

Updated.

**P2L02 lowercase "N"itrogen**

Updated.

**P4L27 provide the reference for validation again.**

Updated.

**P4L31 knowns->known**

Updated.

**P5 l20 space before "."**

Updated.

**Sect 3.2 : Please provide an explicit definition of "hypoxic area". For instance P6L15 "area covered", means area where hypoxia prevails for more than 24h? ( deduced from axis label of Fig 5)**

Updated: "… Any further increase in catchment DIN or TP load had no obvious impact on the size of the total hypoxic area (the area with 24-hour-averaged bottom-water DO concentration < 2mg/L), while slight increases were seen for the other three scenarios. This was because TPP did not increase much when either DIN or TP increased."

**P6 L18 : Why is DON mentioned here. Sect. 2.2 precise DON and DOC are not represented in the model?**

Changed to PON.

**P7 L9-10 : "The ratio ..41)" : I don't understand this sentence. What do you mean. This ratio was 33% instead of 8.5 % at Lake King. What is the $R^2$ referring to ? Please rephrase**

Updated. This is the trend line plotting TPP vs TCO2: " To compare the effect of nutrient reductions on TCO2 fluxes, the effect of the initial sediment nutrient condition was taken into account by subtracting the $TCO_2$ flux and TPP for the simulation with no catchment nutrient input from all the model results. The $TCO_2$ flux and TPP were highly correlated and $TCO_2$ flux was approximately equivalent to 8.5% (calculated by linear regression: $R^2=0.97$, n=41) of the TPP across the entire lake system. The ratio was much higher at Lake King and increased to 33% ($R^2=0.93$, n=41)."

**P7 L21-22 . In the case were all scenario starts from the same ICs (see main question 1), could it also be due to an ongoing mineralization of refractory sediment stocks ?**

Strictly speaking, yes, the ongoing mineralization of refractory sediment stocks contribute majority of the SOD for the zero catchment scenario. This SOD was small but relatively constant over time. However, it was really stratification that prevented oxygen replenishment and induced hypoxia.

**P8 L5: Those "mechanisms" were not mentioned in the results, nor are they clearly described in the following. Clarify or remove this sentence.**

Updated: "…Surprisingly, the model showed that primary production was equally sensitive to inorganic and particulate nitrogen loading and that there were two distinct mechanisms by which these two nitrogen forms were trapped within the lakes. Particulate nitrogen can settle down to the sediment while only a negligible portion of the inorganic nitrogen can be transported to the sediment by diffusion unless converted to particulate form by photosynthesis…"

**P8L6 " the model simulated the transport " -> "we used the model to simulate the transport .. "**

Updated.

**P8 "land use" -> could you expand a bit the discussion here ?**

Updated: "…We therefore suggest further studies need to be undertaken to better understand the degradation kinetics of PON and the factors that control this such as land use which may generate PON of different degradability…"

**P8 L15 : were all biogeochemical processes disabled to estimate transport or only plankton uptake ?**

Updated: "…On the other hand, without any biogeochemical processes such as phytoplankton uptake, only 32% of the DIN remained in the lakes after the flood…"

*P8 L16: Please precise how is the 70% computed and to what the R$^2$ refer.*

This was calculated by plot TN remaining vs TN input, 70% is the trend line with a slope of 0.7. See changes for P8L15-17 below.

*P8 L15-17 : Please rephrase. The reader can understand the message with the next sentence but it is not clear in the present form*

Updated: "…On the other hand, without any biogeochemical processes such as phytoplankton uptake, only 32% of the DIN remained in the lakes after the flood. In fact, all the simulations except for the TP reduction scenario showed that around 70% (calculated by linear regression: R$^2$=0.99, n=37) of TN contributed by the flood event still remained in the lakes by the end of August 2011…"

*P8L18 : please provide explicitly the definition for TN export rate.*

Updated. "…TN exported to the ocean as a percentage of the total catchment nitrogen input increased by 2.3%, 8.1%, 24% and 50% correspondingly when TP was reduced by 25%, 50%, 75% and 100%,…"

*P9 L 16 and following: Might be rephrased for clarity. For instance using the au-tochtonous/allochtonousnomenclature.*

We have reworded this by using 'internal primary production' and 'external catchment input': "There has been controversy as to whether internal primary production  simulated directly by anthropogenic nutrients or external catchment organic carbon inputs caused hypoxia in estuaries such as the Gulf of Mexico (Boesch et al., 2009)…"

*P 10 L8 .. contribute "by" less than 7% "to" the ..  P10 L9 bottom water "Oxygen" depletion*

Updated: "…In addition, the catchment POC only contributed less than 7% to the sediment TCO$_2$ flux between September 2011 and Jan 2012…"

*P10 L23-25, please clarify or better integrate in the current discussion.*

It has been deleted to avoid confusion.

*Fig 5 -> reallocate the definition given in the axis label ( min 24h ..)  to the caption or the text (or both)*

Updated. The definition was moved to the text.

[Figure]

*Fig 6 is maybe not essential, and could be described in words.*

Figure 6 has been deleted.

*Fig8 caption mentions again " and occurence of hypoxia .. of". Is that a Typo ?*

Updated

**Response to the 2nd Reviewer's comments**

1) *Page 3, Line 29 – the sentence that begins here and describes the turbulence closure schemes is organized and worded in an odd way – please re-*

   We have expanded the model description section and made it more readable.
   "…The model consists of two components, the hydrodynamic model and the biogeochemical/ecological model. The hydrodynamic model simulated the transport and turbulent mixing in the water column…"

2) *A model of bio-irrigation is mentioned (Page 4, Line 15) without any real description of it. I realize there is another paper describing this model, and after reading that paper (Zhu et al. 2016) I am convinced that this bio-irrigation model could be described in the current paper in just a few sentences to provide the basic details.*

   We have expanded the model description section and made it more readable. See point 4 below.

3) *Page 4, Lines 3-5 – has the wave model been validated at all? It is not in this paper, and from what I can tell, was not validated in Zhu et al. (2016). Either some validation is due in this paper, or the authors should cite where the validation exists.*

   The wave model was not validated and we have added a sentence to explain it. "…. Since there was not any measured wave data inside the lakes, the default wave input parameters was used…"

4) *Page 4, Lines 23-24: A" spatially-varying sediment iron-bound phosphate. . .." Is mentioned as an initial condition, but it is not clear at all how iron-phosphorus dynamics are represented by the model, and these dynamics are not really described in Zhu et al. (2016). I assume you are not modeling iron explicitly, but rather that there is an adsorption between iron-bound and free phosphate that is modulated by oxygen and assumes unlimited adsorption (Zhu et al. 2016 include an adsorption parameter in the appendix table). You could clarify this with two additional sentences I think. For this comment and comment #2, I believe strongly that a given paper should attempt to be a stand-alone document, and cannot completely rely on a previous paper to describe the model. Obviously you cannot re-write the entire model description, but each model component that is highlighted should have a basic description of it and the original source of the details.*

   We have expanded the model description section and made it more readable. Yes, iron was not modelled explicitly but treated as a spatially varying constant estimated by previous studies: "…The present model overcame previous limitations by implementing sorption and desorption of sediment phosphate, and bioirrigation into the model enabling an accurate simulation of

sediment phosphorus dynamics. The sorption and desorption of sediment phosphate were modelled explicitly based on the penetration depth of oxygen and nitrate, and the sediment iron concentration which was a spatially varying constant estimated by using the data collected from previous studies. The impact of bioirrigation was modelled by introducing a scaling factor that was used to adjust the diffusion rates of oxygen and inorganic nutrient at the sediment-water interface. The scaling factor was a function of temperature, DO and labile organic matter…."

5) **There are a fair amount of basic grammatical errors in the text – please read over carefully to correct this.**

We have conducted thorough editing and proofreading.

6) **Page 7, Line 9: I assume that TPP is gross photosynthesis. If so, while it is relevant to relate TPP to the external POC loading, I think it is less useful for comparing TPP to sediment CO2 fluxes. I think it would be more appropriate to relate TPP-R (net phytoplankton production) to sediment CO2 fluxes, because it is the net production that potentially yields carbon that can sink to sediments to support CO2 production.**

Yes, TPP is the gross photosynthesis. It is relatively easy to track and extract from the model. However, the net phytoplankton production is not a readily available output. On the other hand, TPP-R is strongly correlated to TPP. Therefore we consider the use of TPP in current study was also appropriate.

7) **Page 7, Line 17: Please define explicitly how you computed denitrification efficiency.**

The definition of denitrification efficiency was included: "…The denitrification efficiency here has been defined as the percentage of inorganic nitrogen released from the sediment as dinitrogen gas (g N/m$^2$/year) and can be calculated by $[N_2/(FNH + FN3 + N_2) \times 100\%]$ (Eyre and Ferguson, 2009). FNH and FN3 are the sediment ammonia and nitrate fluxes in g N /m$^2$/year…"

8) **Page 9, Lines 6-8: Please provide values for bottom shear stress to help the reader understand what "very low" means, relative to the rest of Gippsland lakes and other systems.**

Updated: "…Another important reason for POC retention in the Lake King basin was that the bottom shear stress in this area was generally low and the 90$^{th}$ percentile shear velocity was only 0.34 cm/s which was lower than the reported critical shear velocity (0.4-0.8 cm/s) for the resuspension of phytoplankton-derived organic matter (Beaulieu, 2003)…"

9) **Page 10, Line 23-24: I think it is worth stating clearly what the mechanism is that limits internal phosphorus loading with elevated nitrate. Although you cite literature, the mechanism is not intuitive and perhaps not widely known.**

In line 25, we stated that: the model showed that the increase in oxidised depth of the sediment was limited due to increased sediment oxygen demand and low diffusion rate of nitrate in the sediment. This sentence has been deleted, as it does not fit well in the context of the discussion as what has been recommended by the 1st reviewer.

10) **Page 11, Lines 4-5: What do you mean by the sentence "However, initial input of catchment phosphorus. . .. . ."? What analysis or model run is this based on? There are not analyses in the paper to support this, and in the absence of such a statement, this sentence appears to contradict the one that came just before it**

We came to this conclusion by comparing the zero TP scenario to the other ones. If there is no TP from the catchment, there would be no primary production. There must be P input for primary production to trap the N, especially IN. This P would likely be contributed from the catchment, because the sediment P release would not occur until bottom oxygen was depleted.

11) **On Page 10, Lines 9-10, you state that bottom oxygen depletion in Lake King is primarily related to nutrient inputs and phytoplankton production, and your scenarios indicate that elevated TP loads did little to stimulate additional TPP and hypoxia. But then on Page 11, Lines 12-15, you indicate that there could be a recalcitrance of the system in the face of modest nutrient reductions due to internal phosphorus loading from sediment stores, which seems to contradict the prior statements. Please clarify the specifics of this in the manuscript if that is possible, as it leaves the reader wanting for a resolution.**

This was because sediment P rather than the catchment P supplied majority of the P to support TPP and hypoxia. The sediment P originated from the catchment. Therefore, reduction in catchment P will eventually reduce the P accumulation in the sediment.

12) **You also indicate a 5-10 year time frame for the exhaustion of internal P stores, but what is that based upon? It would seem easy to cut off new nutrient inputs and re- run the model for 5-10 more years of no-new nutrient loads and quantify for how many years the sediments continue to release phosphorus without new watershed inputs**

This statement was based on the long-term studies of some European lakes (Schindler and Hecky, 2009), the 5-10 year was an expectation only. We have removed this statement to avoid confusion.
We acknowledge the reviewer's comment that we would also like to do a long term simulation; but the model require intensive computation and more importantly good long term data. We will work on this and hopefully address this in a separate manuscript.

13) **The paragraph ending Page 11 needs grammatical editing.**

We have conducted thorough editing and proofreading.

14) **Conclusion section, Page 12, lines 1-3: The first two statements of this paragraph, again, appear to contradict one another. The first sentence says hypoxia is driven by stratification and sediment carbon enrichment, while the second says nitrogen- stimulated primary production was responsible for DO depletion. So which is it? Again, why would the internal phosphorus loading matter if nitrogen is the key limiting nutrient? You did show that TP loading increases stimulated TPP beyond what TN and POC stimulated – so perhaps some improved wording would help.**

Updated. We should say that the carbon enrichment was caused by primary production. In the Gippsland Lakes, either N or P could be the limiting nutrient depending on catchment input and sediment geochemistry, although we found N was typically the limiting one. On the other hand, both field studies and model results showed that it was sediment P release that supported the summer blooms. In fact, Figure 4 showed that increases in TP loading were not as effective as the increase TN in terms of stimulating TPP.

15) **Figure 2: I think it would be easier to see the flow record if it had its own panel**

Figure 2 was updated.

[Figure]

16) *Units: I understand the value of using tons to represent large numbers, but it might also help to indicate, perhaps in the text, what the sediment-water ammonium and phosphate fluxes were in commonly used units (micromole/m-2/h-1). Perhaps simply contrasting the rates at the highest nutrient increase and larges reduction. This would help compare these numbers with other systems.*

We have added a second y axis using mmole/m2/day for the CO2 and nutrient fluxes in figure 8,9 and 11 which can help the reader who would like to compare the numbers with the molar units.

[revised manuscript text omitted]

MONBET, P., MCKELVIE, I. D. & WORSFOLD, P. J. 2007. Phosphorus speciation, burial and regeneration in coastal lagoon sediments of the Gippsland Lakes (Victoria, Australia). *Environmental Chemistry,* 4**,** 334-346.

MULHOLLAND, P. J., HELTON, A. M., POOLE, G. C., HALL, R. O., HAMILTON, S. K., PETERSON, B. J., TANK, J. L., ASHKENAS, L. R., COOPER, L. W. & DAHM, C. N. 2008. Stream denitrification across biomes and its response to anthropogenic nitrate loading. *Nature,* 452**,** 202-205.

[revised manuscript text omitted]

---

## Author Response (AR2)

MONASH University

28 August, 2017

Professor Caroline P. Slomp
Handling Associate Editor
Biogeosciences
Princetonplein 9
Room 1.08
3584 CC  UTRECHT
The Netherlands

Dear Professor Slomp,

Please find attached revision of the manuscript bg-2017-132, entitled "Effects of changes in nutrient loading and composition on hypoxia dynamics and internal nutrient cycling of a stratified coastal lagoon". We have carefully considered the comments from the first reviewer and undertaken the suggested changes as summarised below in this document.

We hope the manuscript can now meet your expectations for publication and if you have any further requirements we would be pleased to make any necessary changes.

Yours Sincerely

Yafei Zhu

**Response to the 1st Reviewer's comments**

*\* Although it has been improved, the benthic model component has not yet been sketched clearly enough. Please precise its structure, and mention burial in the description. I understand a complete and detailed description is too heavy, but some aspects strongly constrain the nature and reach of this manuscript's conclusions (ie. the range or classes of lability considered for benthic organic matter, the possibility for burial, the possibility for delayed resuspension, etc..). In answer to our comments regarding burial, the author mention the "total depth", without definition. I assume it refers to thickness of the accumulated sediment layer, but this is unsure.  In any case, any precision has to be done in the manuscript for readers, not just as answers to the reviewers.  The following sentence is ill formulated: " An accumulation of organic matter layer at the floor was formed due to settling. ". The 2-layer topology of the benthic component comes last in the current description, but should come first.*

We added the following to describe the burial process of organic matter: "Burial was active when the total thickness of sediment organic matter exceeded 20 cm.  The thickness was calculated using a density of 16.959kg C/m$^3$ assuming a porosity of 0.8.   The buried sediment organic matter was removed from simulations."

Sorry for the confusion, but it was not really 2-layer topology for the benthic component, more precisely it should be "An accumulation of organic matter layer in the bottom water was formed due to settling. ". This layers is still moving with water and dispersible just like other parts of the water column.

*\* The fact that sediment P stocks last longer than N is a consequence of modelling choices : the range of lablity considered and the presence of iron-bound P stocks.*
*Bearing several consequences in the study conclusions, the implications of those modelling choices should be stated more clearly (it adds to the already included conclusions regarding the need for a better description of OM lability).*

Sediment P stocks last longer than N is found in many systems due to physical and chemical characteristics of P (ie. Adsorption/desorption of P; unlike nitrogen, there was no removal process for P apart from burial).  We consider the presence of iron-bound P in the Gippsland Lakes is also a fact rather than a modelling choice. The range of lability used would not have any impact on the fact that large quantity of iron-bound phosphorus is stored in the sediment and the adsorption/desorption process phosphorus undertakes.

*\* Rephrase P6L2 ".. and this confirmed that 60% of the particulate nutrient were labile was a reasonable assumption. "*

Rephrased: …and this confirmed that 60% of the particulate nutrient were labile was a reasonable assumption to describe the characteristics of catchment organic nutrient loads.

*Rephrase P7L7 "On the other hand, the hypoxic area reduced more steadily when TN was reduced and decreased the most when TN and TP were reduced."*

Rephrased: On the other hand, the hypoxic area reduced more steadily when TN was reduced and the magnitude of decrease in the hypoxic area was most obvious when  both TN and TP were reduced

*P7L27 : " .. the deposition rate in Lake King was much higher than IN the rest of the lakes .. "*

"in" was added to the sentences

*The definition of total hypoxic area remains unclear.*
*The proposed definition ("the area with 24-hour-averaged bottom-water DO concentration < 2mg/L") defines a value that changes in time. However, Fig 5. and the corresponding discussion consider a single number per simulation to characterize the hypoxic area. Is the "total hypoxic area" the area where hypoxia (as defined above) occurs at least once during the simulation ?  Or the area where hypoxia occurs all the time ? Or the maximum hypoxic area achieved during the simulation (ie, area simultaneously covered by hypoxia) ? Please precise the definition.*

The definition is updated: the total area with 24-hour-averaged bottom-water DO concentration < 2mg/L that occurred at least once

*P10L7 : "simulated' -> "stimulated"*

"Simulated" was changed to "stimulated"

*P10L42 IF applying -> By Applying*

"IF applying" was changed to "By Applying"

[revised manuscript text omitted]